# Learning Mixed Multinomial Logit Model from Ordinal Data

**Sewoong Oh**
Dept. of Industrial and Enterprise Systems Engr.
University of Illinois at Urbana-Champaign
Urbana, IL 61801
swoh@illinois.edu

**Devavrat Shah**
Department of Electrical Engineering
Massachussetts Institute of Technology
Cambridge, MA 02139
devavrat@mit.edu

## Abstract

Motivated by generating personalized recommendations using ordinal (or preference) data, we study the question of learning a mixture of MultiNomial Logit (MNL) model, a parameterized class of distributions over permutations, from partial ordinal or preference data (e.g. pair-wise comparisons). Despite its long standing importance across disciplines including social choice, operations research and revenue management, little is known about this question. In case of single MNL models (no mixture), computationally and statistically tractable learning from pair-wise comparisons is feasible. However, even learning mixture with two MNL components is infeasible in general.

Given this state of affairs, we seek conditions under which it is feasible to learn the mixture model in both computationally and statistically efficient manner. We present a sufficient condition as well as an efficient algorithm for learning mixed MNL models from partial preferences/comparisons data. In particular, a mixture of $r$ MNL components over $n$ objects can be learnt using samples whose size scales polynomially in $n$ and $r$ (concretely, $r^{3.5}n^3(\log n)^4$, with $r \ll n^{2/7}$ when the model parameters are sufficiently *incoherent*). The algorithm has two phases: first, learn the pair-wise marginals for each component using tensor decomposition; second, learn the model parameters for each component using RANKCEN-TRALITY introduced by Negahban et al. In the process of proving these results, we obtain a generalization of existing analysis for tensor decomposition to a more realistic regime where only partial information about each sample is available.

## 1 Introduction

**Background.** Popular recommendation systems such as collaborative filtering are based on a partially observed *ratings* matrix. The underlying hypothesis is that the true/latent score matrix is low-rank and we observe its partial, noisy version. Therefore, matrix completion algorithms are used for learning, cf. [8, 14, 15, 20]. In reality, however, observed preference data is not just scores. For example, clicking one of the many choices while browsing provides partial order between clicked choice versus other choices. Further, scores do convey ordinal information as well, e.g. score of 4 for paper A and score of 7 for paper B by a reviewer suggests ordering B > A. Similar motivations led Samuelson to propose the *Axiom of revealed preference* [21] as the model for rational behavior. In a nutshell, it states that *consumers have latent order of all objects, and the revealed preferences through actions/choices are consistent with this order.* If indeed all consumers had identical ordering, then learning preference from partial preferences is effectively the question of sorting.

In practice, individuals have different orderings of interest, and further, each individual is likely to make noisy choices. This naturally suggests the following model – each individual has a latent distribution over orderings of objects of interest, and the revealed partial preferences are consistent

with it, i.e. samples from the distribution. Subsequently, the preference of the population as a whole can be associated with a distribution over permutations. Recall that the low-rank structure for score matrices, as a model, tries to capture the fact that there are only a few different types of choice profile. In the context of modeling consumer choices as distribution over permutation, MultiNomial Logit (MNL) model with a small number of mixture components provides such a model.

**Mixed MNL.** Given $n$ objects or choices of interest, an MNL model is described as a parametric distribution over permutations of $n$ with parameters $\mathbf{w} = [w_i] \in \mathbb{R}^n$: each object $i$, $1 \le i \le n$, has a parameter $w_i > 0$ associated with it. Then the permutations are generated randomly as follows: choose one of the $n$ objects to be ranked 1 at random, where object $i$ is chosen to be ranked 1 with probability $w_i / (\sum_{j=1}^n w_j)$. Let $i_1$ be object chosen for the first position. Now to select second ranked object, choose from remaining with probability proportional to their weight. We repeat until all objects for all ranked positions are chosen. It can be easily seen that, as per this model, an item $i$ is ranked higher than $j$ with probability $w_i / (w_i + w_j)$.

In the mixed MNL model with $r \ge 2$ mixture components, each component corresponds to a different MNL model: let $\mathbf{w}^{(1)}, \ldots, \mathbf{w}^{(r)}$ be the corresponding parameters of the $r$ components. Let $\mathbf{q} = [q_a] \in [0,1]^r$ denote the mixture distribution, i.e. $\sum_a q_a = 1$. To generate a permutation at random, first choose a component $a \in \{1, \ldots, r\}$ with probability $q_a$, and then draw random permutation as per MNL with parameters $\mathbf{w}^{(a)}$.

**Brief history.** The MNL model is an instance of a class of models introduced by Thurstone [23]. The description of the MNL provided here was formally established by McFadden [17]. The same model (in form of pair-wise marginals) was introduced by Zermelo [25] as well as Bradley and Terry [7] independently. In [16], Luce established that MNL is the only distribution over permutation that satisfies the axiom of Independence from Irrelevant Alternatives.

On learning distributions over permutations, the question of learning single MNL model and more generally instances of Thurstone's model have been of interest for quite a while now. The maximum likelihood estimator, which is logistic regression for MNL, has been known to be consistent in large sample limit, cf. [13]. Recently, RANKCENTRALITY [19] was established to be statistical efficient. For learning sparse mixture model, i.e. distribution over permutations with each mixture being delta distribution, [11] provided sufficient conditions under which mixtures can be learnt exactly using pair-wise marginals – effectively, as long as the number of components scaled as $o(\log n)$ where components satisfied appropriate *incoherence* condition, a simple iterative algorithm could recover the mixture. However, it is not robust with respect to noise in data or finite sample error in marginal estimation. Other approaches have been proposed to recover model using convex optimization based techniques, cf. [10, 18]. MNL model is a special case of a larger family of discrete choice models known as the Random Utility Model (RUM), and an efficient algorithm to learn RUM is introduced in [22]. Efficient algorithms for learning RUMs from partial rankings has been introduced in [3, 4]. We note that the above list of references is very limited, including only closely related literature. Given the nature of the topic, there are a lot of exciting lines of research done over the past century and we shall not be able to provide comprehensive coverage due to a space limitation.

**Problem.** Given observations from the mixed MNL, we wish to learn the model parameters, the mixing distribution $\mathbf{q}$, and parameters of each component $\mathbf{w}^{(1)}, \ldots, \mathbf{w}^{(r)}$. The observations are in form of pair-wise comparisons. Formally, to generate an observation, first one of the $r$ mixture components is chosen; and then for $\ell$ of all possible $\binom{n}{2}$ pairs, comparison outcome is observed as per this MNL component[1]. These $\ell$ pairs are chosen, uniformly at random, from a pre-determined $N \le \binom{n}{2}$ pairs: $\{(i_k, j_k), 1 \le k \le N\}$. We shall assume that the selection of $N$ is such that the undirected graph $G = ([n], E)$, where $E = \{(i_k, j_k) : 1 \le k \le N\}$, is connected.

We ask following questions of interest: Is it always feasible to learn mixed MNL? If not, under what conditions and how many samples are needed? How computationally expensive are the algorithms?

We briefly recall a recent result [1] that suggests that it is impossible to learn mixed MNL models in general. One such example is described in Figure 1. It depicts an example with $n = 4$ and $r = 2$ and a uniform mixture distribution. For the first case, in mixture component 1, with probability 1 the ordering is $a > b > c > d$ (we denote $n = 4$ objects by $a, b, c$ and $d$); and in mixture component 2, with probability 1 the ordering is $b > a > d > c$. Similarly for the second case, the two mixtures are made up of permutations $b > a > c > d$ and $a > b > d > c$. It is easy to see the distribution over any 3-wise comparisons generated from these two mixture models is identical. Therefore, it is impossible to differentiate these two using 3-wise or pair-wise comparisons. In general, [1] established that there exist mixture distributions with $r \leq n/2$ over $n$ objects that are impossible to distinguish using $\log n$-wise comparison data. That is, learning mixed MNL is not always possible.

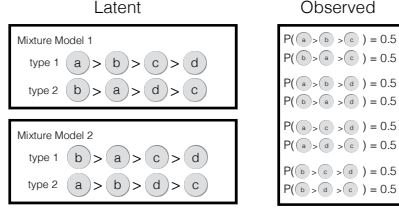

Figure 1: Two mixture models that cannot be differentiated even with 3-wise preference data.

**Contributions.** The main contribution of this work is identification of sufficient conditions under which mixed MNL model can be learnt efficiently, both statistically and computationally. Concretely, we propose a two-phase learning algorithm: in the first phase, using a tensor decomposition method for learning mixture of discrete product distribution, we identify pair-wise marginals associated with each of the mixture; in the second phase, we use these pair-wise marginals associated with each mixture to learn the parameters associated with each of the MNL mixture component.

The algorithm in the first phase builds upon the recent work by Jain and Oh [12]. In particular, Theorem 3 generalizes their work for the setting where for each sample, we have limited information - as per [12], we would require that each individual gives the entire permutation; instead, we have extended the result to be able to cope with the current setting when we only have information about $\ell$, potentially finite, pair-wise comparisons. The algorithm in the second phase utilizes RANK-CENTRALITY [19]. Its analysis in Theorem 4 works for setting where observations are no longer independent, as required in [19].

We find that as long as certain rank and *incoherence* conditions are satisfied by the parameters of each of the mixture, the above described two phase algorithm is able to learn mixture distribution $\mathbf{q}$ and parameters associated with each mixture, $\mathbf{w}^{(1)}, \ldots, \mathbf{w}^{(r)}$ faithfully using samples that scale polynomially in $n$ and $r$ – concretely, the number of samples required scale as $r^{3.5} n^3 (\log n)^4$ with constants dependent on the incoherence between mixture components, and as long as $r \ll n^{2/7}$ as well as $G$, the graph of potential comparisons, is a spectral expander with the total number of edges scaling as $N = O(n \log n)$. For the precise statement, we refer to Theorem 1.

The algorithms proposed are iterative, and primarily based on spectral properties of underlying tensors/matrices with provable, fast convergence guarantees. That is, algorithms are not only polynomial time, they are practical enough to be scalable for high dimensional data sets.

**Notations.** We use $[N] = \{1, \ldots, N\}$ for the first $N$ positive integers. We use $\otimes$ to denote the outer product such that $(x \otimes y \otimes z)_{ijk} = x_i y_j z_k$. Given a third order tensor $T \in \mathbb{R}^{n_1 \times n_2 \times n_3}$ and a matrix $U \in \mathbb{R}^{n_1 \times r_1}, V \in \mathbb{R}^{n_2 \times r_2}, W \in \mathbb{R}^{n_3 \times r_3}$, we define a linear mapping $T[U, V, W] \in \mathbb{R}^{r_1 \times r_2 \times r_3}$ as $T[U, V, W]_{abc} = \sum_{i,j,k} T_{ijk} U_{ia} V_{jb} W_{kc}$. We let $\|x\| = \sqrt{\sum_i x_i^2}$ be the Euclidean norm of a vector, $\|M\|_2 = \max_{\|x\| \leq 1, \|y\| \leq 1} x^T M y$ be the operator norm of a matrix, and $\|M\|_F = \sqrt{\sum_{i,j} M_{ij}^2}$ be the Frobenius norm. We say an event happens with high probability (w.h.p) if the probability is lower bounded by $1 - f(n)$ such that $f(n) = o(1)$ as $n$ scales to $\infty$.

## 2  Main result

In this section, we describe the main result: sufficient conditions under which mixed MNL models can be learnt using tractable algorithms. We provide a useful illustration of the result as well as discuss its implications.

**Definitions.** Let $\mathcal{S}$ denote the collection of observations, each of which is denoted as $N$ dimensional, $\{-1, 0, +1\}$ valued vector. Recall that each observation is obtained by first selecting one of the $r$ mixture MNL component, and then viewing outcomes, as per the chosen MNL mixture component, of $\ell$ randomly chosen pair-wise comparisons from the $N$ pre-determined comparisons $\{(i_k, j_k) : 1 \leq i_k \neq j_k \leq n, 1 \leq k \leq N\}$. Let $x_t \in \{-1, 0, +1\}^N$ denote the $t$th observation with $x_{t,k} = 0$ if the $k$th pair $(i_k, j_k)$ is not chosen amongst the $\ell$ randomly chosen pairs, and $x_{t,k} = +1$ (respectively $-1$) if $i_k < j_k$ (respectively $i_k > j_k$) as per the chosen MNL mixture component. By definition, it is easy to see that for any $t \in \mathcal{S}$ and $1 \leq k \leq N$,

$$\mathbb{E}[x_{t,k}] = \frac{\ell}{N}\Big[\sum_{a=1}^{r} q_a P_{ka}\Big], \text{ where } P_{ka} = \frac{w_{j_k}^{(a)} - w_{i_k}^{(a)}}{w_{j_k}^{(a)} + w_{i_k}^{(a)}}. \tag{1}$$

We shall denote $P_a = [P_{ka}] \in [-1, 1]^N$ for $1 \leq a \leq r$. Therefore, in a vector form

$$\mathbb{E}[x_t] = \frac{\ell}{N} P\mathbf{q}, \text{ where } P = [P_1 \ldots P_r] \in [-1, 1]^{N \times r}. \tag{2}$$

That is, $P$ is a matrix with $r$ columns, each representing one of the $r$ mixture components and $\mathbf{q}$ is the mixture probability. By independence, for any $t \in \mathcal{S}$, and any two different pairs $1 \leq k \neq m \leq N$,

$$\mathbb{E}[x_{t,k} x_{t,m}] = \frac{\ell^2}{N^2}\Big[\sum_{a=1}^{r} q_a P_{ka} P_{ma}\Big]. \tag{3}$$

Therefore, the $N \times N$ matrix $\mathbb{E}[x_t x_t^T]$ or equivalently tensor $\mathbb{E}[x_t \otimes x_t]$ is proportional to $M_2$ except in diagonal entries, where

$$M_2 = PQP^T \equiv \sum_{a=1}^{r} q_a (P_a \otimes P_a), \tag{4}$$

$Q = \text{diag}(\mathbf{q})$ being diagonal matrix with its entries being mixture probabilities, $\mathbf{q}$. In a similar manner, the tensor $\mathbb{E}[x_t \otimes x_t \otimes x_t]$ is proportional to $M_3$ (except in $O(N^2)$ entries), where

$$M_3 = \sum_{a=1}^{r} q_a (P_a \otimes P_a \otimes P_a). \tag{5}$$

Indeed, empirical estimates $\hat{M}_2$ and $\hat{M}_3$, defined as

$$\hat{M}_2 = \frac{1}{|\mathcal{S}|}\Big[\sum_{t \in \mathcal{S}} x_t \otimes x_t\Big], \text{ and } \hat{M}_3 = \frac{1}{|\mathcal{S}|}\Big[\sum_{t \in \mathcal{S}} x_t \otimes x_t \otimes x_t\Big], \tag{6}$$

provide good proxy for $M_2$ and $M_3$ for large enough number of samples; and shall be utilized crucially for learning model parameters from observations.

**Sufficient conditions for learning.** With the above discussion, we state sufficient conditions for learning the mixed MNL in terms of properties of $M_2$:

C1. $M_2$ has rank $r$; let $\sigma_1(M_2)$, $\sigma_r(M_2) > 0$ be the largest and smallest singular values of $M_2$.

C2. For a large enough universal constant $C' > 0$,

$$N \geq C' r^{3.5} \mu^6(M_2) \Big(\frac{\sigma_1(M_2)}{\sigma_r(M_2)}\Big)^{4.5}. \tag{7}$$

In the above, $\mu(M_2)$ represents incoherence of a symmetric matrix $M_2$. We recall that for a symmetric matrix $M \in \mathbb{R}^{N \times N}$ of rank $r$ with singular value decomposition $M = USU^T$, the incoherence is defined as

$$\mu(M) = \sqrt{\frac{N}{r}} \Big(\max_{i \in [N]} \|U_i\|\Big). \tag{8}$$

C3. The undirected graph $G = ([n], E)$ with $E = \{(i_k, j_k) : 1 \leq k \leq N\}$ is connected. Let $A \in \{0, 1\}^{n \times n}$ be adjacency matrix with $A_{ij} = 1$ if $(i, j) \in E$ and $0$ otherwise; let $D = \text{diag}(d_i)$ with $d_i$ being degree of vertex $i \in [n]$ and let $L_G = D^{-1}A$ be normalized Laplacian of $G$. Let $d_{\max} = \max_i d_i$ and $d_{\min} = \min_i d_i$. Let the $n$ eigenvalues of stochastic matrix $L_G$ be $1 = \lambda_1(L_G) \geq \ldots \lambda_n(L_G) \geq -1$. Define spectral gap of $G$:

$$\xi(G) = 1 - \max\{\lambda_2(L), -\lambda_n(L)\}. \tag{9}$$

Note that we choose a graph $G = ([n], E)$ to collect pairwise data on, and we want to use a graph that is connected, has a large spectral gap, and has a small number of edges. In condition (C3), we need connectivity since we cannot estimate the relative strength between disconnected components (e.g. see [13]). Further, it is easy to generate a graph with spectral gap $\xi(G)$ bounded below by a universal constant (e.g. $1/100$) and the number of edges $N = O(n \log n)$, for example using the configuration model for Erdös-Renyi graphs. In condition (C2), we require the matrix $M_2$ to be sufficiently incoherent with bounded $\sigma_1(M_2)/\sigma_r(M_2)$. For example, if $q_{\max}/q_{\min} = O(1)$ and the profile of each type in the mixture distribution is sufficiently different, i.e. $\langle P_a, P_b \rangle / (\|P_a\|\|P_b\|) < 1/(2r)$, then we have $\mu(M_2) = O(1)$ and $\sigma_1(M_2)/\sigma_r(M_2) = O(1)$. We define $b = \max_{a=1}^r \max_{i,j \in [n]} w_i^{(a)}/w_j^{(a)}$, $q_{\max} = \max_a q_a$, and $q_{\min} = \min_a q_a$. The following theorem provides a bound on the error and we refer to the appendix for a proof.

**Theorem 1.** *Consider a mixed MNL model satisfying conditions (C1)-(C3). Then for any $\delta \in (0,1)$, there exists positive numerical constants $C, C'$ such that for any positive $\varepsilon$ satisfying*

$$0 < \varepsilon < \Big( \frac{q_{\min} \xi^2(G) d_{\min}^2}{16 q_{\max} r \, \sigma_1(M_2) b^5 d_{\max}^2} \Big)^{0.5}, \tag{10}$$

*Algorithm 1 produces estimates $\hat{\mathbf{q}} = [\hat{q}_a]$ and $\hat{\mathbf{w}} = [\hat{\mathbf{w}}^{(a)}]$ so that with probability at least $1 - \delta$,*

$$\big| \, \hat{q}_a - q_a \, \big| \le \varepsilon, \text{ and}$$

$$\frac{\|\hat{\mathbf{w}}^{(a)} - \mathbf{w}^{(a)}\|}{\|\mathbf{w}^{(a)}\|} \le C \Big( \frac{r \, q_{\max} \sigma_1(M_2) b^5 d_{\max}^2}{q_{\min} \xi^2(G) d_{\min}^2} \Big)^{0.5} \varepsilon, \tag{11}$$

*for all $a \in [r]$, as long as*

$$|\mathcal{S}| \ge C' \frac{r N^4 \log(N/\delta)}{q_{\min} \sigma_1(M_2)^2 \varepsilon^2} \Big( \frac{1}{\ell^2} + \frac{\sigma_1(M_2)}{\ell N} + \frac{r^4 \sigma_1(M_2)^4}{\sigma_r(M_2)^5} \Big). \tag{12}$$

**An illustration of Theorem 1.** To understand the applicability of Theorem 1, consider a concrete example with $r = 2$; let the corresponding weights $\mathbf{w}^{(1)}$ and $\mathbf{w}^{(2)}$ be generated by choosing each weight uniformly from $[1, 2]$. In particular, the rank order for each component is a uniformly random permutation. Let the mixture distribution be uniform as well, i.e. $\mathbf{q} = [0.5 \ 0.5]$. Finally, let the graph $G = ([n], E)$ be chosen as per the Erdös-Rényi model with each edge chosen to be part of the graph with probability $\bar{d}/n$, where $\bar{d} > \log n$. For this example, it can be checked that Theorem 1 guarantees that for $\varepsilon \le C/\sqrt{n\bar{d}}$, $|\mathcal{S}| \ge C' n^2 \bar{d}^2 \log(n\bar{d}/\delta)/(\ell \varepsilon^2)$, and $n\bar{d} \ge C'$, we have for all $a \in \{1, 2\}$, $|\hat{q}_a - q_a| \le \varepsilon$ and $\|\hat{\mathbf{w}}^{(a)} - \mathbf{w}^{(a)}\|/\|\mathbf{w}^{(a)}\| \le C'' \sqrt{n\bar{d}} \varepsilon$. That is, for $\ell = \Theta(1)$ and choosing $\varepsilon = \varepsilon'/(\sqrt{n\bar{d}})$, we need sample size of $|\mathcal{S}| = O(n^3 \bar{d}^3 \log n)$ to guarantee error in both $\hat{\mathbf{q}}$ and $\hat{\mathbf{w}}$ smaller than $\varepsilon'$. Instead, if we choose $\ell = \Theta(n\bar{d})$, we only need $|\mathcal{S}| = O((n\bar{d})^2 \log n)$. Limited samples per observation leads to penalty of factor of $(n\bar{d}/\ell)$ in sample complexity. To provide bounds on the problem parameters for this example, we use standard concentration arguments. It is well known for Erdös-Rényi random graphs (see [6]) that, with high probability, the number of edges concentrates in $[(1/2)\bar{d}n, (3/2)\bar{d}n]$ implying $N = \Theta(\bar{d}n)$, and the degrees also concentrate in $[(1/2)\bar{d}, (3/2)\bar{d}]$, implying $d_{\max} = d_{\min} = \Theta(\bar{d})$. Also using standard concentration arguments for spectrum of random matrices, it follows that the spectral gap of $G$ is bounded by $\xi \ge 1 - (C/\sqrt{\bar{d}}) = \Theta(1)$ w.h.p. Since we assume the weights to be in $[1, 2]$, the dynamic range is bounded by $b \le 2$. The following Proposition shows that $\sigma_1(M_2) = \Theta(N) = \Theta(\bar{d}n)$, $\sigma_2(M_2) = \Theta(\bar{d}n)$, and $\mu(M_2) = \Theta(1)$.

**Proposition 2.1.** *For the above example, when $\bar{d} \ge \log n$, $\sigma_1(M_2) \le 0.02N$, $\sigma_2(M_2) \ge 0.017N$, and $\mu(M_2) \le 15$ with high probability.*

Supposen now for general $r$, we are interested in well-behaved scenario where $q_{\max} = \Theta(1/r)$ and $q_{\min} = \Theta(1/r)$. To achieve arbitrary small error rate for $\|\hat{\mathbf{w}}^{(a)} - \mathbf{w}^{(a)}\|/\|\mathbf{w}^{(a)}\|$, we need $\epsilon = O(1/\sqrt{r N})$, which is achieved by sample size $|\mathcal{S}| = O(r^{3.5} n^3 (\log n)^4)$ with $\bar{d} = \log n$.

## 3 Algorithm

We describe the algorithm achieving the bound in Theorem 1. Our approach is two-phased. First, learn the moments for mixtures using a tensor decomposition, cf. Algorithm 2: for each type $a \in [r]$,

produce estimate $\hat{q}_a \in \mathbb{R}$ of the mixture weight $q_a$ and estimate $\hat{P}_a = [\hat{P}_{1a} \ldots \hat{P}_{Na}]^T \in \mathbb{R}^N$ of the expected outcome $P_a = [P_{1a} \ldots P_{Na}]^T$ defined as in (1). Secondly, for each $a$, using the estimate $\hat{P}_a$, apply RANKCENTRALITY, cf. Section 3.2, to estimate $\hat{\mathbf{w}}^{(a)}$ for the MNL weights $\mathbf{w}^{(a)}$.

---

**Algorithm 1**

---

1: **Input**: Samples $\{x_t\}_{t \in \mathcal{S}}$, number of types $r$, number of iterations $T_1, T_2$, graph $G([n], E)$
2: $\{(\hat{q}_a, \hat{P}_a)\}_{a \in [r]} \leftarrow$ SPECTRALDIST $(\{x_t\}_{t \in \mathcal{S}}, r, T_1)$          (see Algorithm 2)
3: **for** $a = 1, \ldots, r$ **do**
4:     set $\tilde{P}_a \leftarrow \mathcal{P}_{[-1,1]}(\hat{P}_a)$ where $\mathcal{P}_{[-1,1]}(\cdot)$ is the projection onto $[-1,1]^N$
5:     $\hat{w}^{(a)} \leftarrow$ RANKCENTRALITY $\left(G, \tilde{P}_a, T_2\right)$          (see Section 3.2)
6: **end for**
7: **Output**: $\{(\hat{q}^{(a)}, \hat{\mathbf{w}}^{(a)})\}_{a \in [r]}$

---

To achieve Theorem 1, $T_1 = \Theta\big( \log(N |\mathcal{S}|) \big)$ and $T_2 = \Theta\big( b^2 d_{\max}(\log n + \log(1/\varepsilon))/(\xi d_{\min}) \big)$ is sufficient. Next, we describe the two phases of algorithms and associated technical results.

### 3.1 Phase 1: Spectral decomposition.

To estimate $P$ and $\mathbf{q}$ from the samples, we shall use tensor decomposition of $\hat{M}_2$ and $\hat{M}_3$, the empirical estimation of $M_2$ and $M_3$ respectively, recall (4)-(6). Let $M_2 = U_{M_2} \Sigma_{M_2} U_{M_2}^T$ be the eigenvalue decomposition and let

$$H = M_3[U_{M_2} \Sigma_{M_2}^{-1/2}, U_{M_2} \Sigma_{M_2}^{-1/2}, U_{M_2} \Sigma_{M_2}^{-1/2}] .$$

The next theorem shows that $M_2$ and $M_3$ are sufficient to learn $P$ and $\mathbf{q}$ exactly, when $M_2$ has rank $r$ (throughout, we assume that $r \ll n \leq N$).

**Theorem 2** (Theorem 3.1 [12]). *Let $M_2 \in \mathbb{R}^{N \times N}$ have rank $r$. Then there exists an orthogonal matrix $V^H = [v_1^H \ v_2^H \ \ldots \ v_r^H] \in \mathbb{R}^{r \times r}$ and eigenvalues $\lambda_a^H$, $1 \leq a \leq r$, such that the orthogonal tensor decomposition of $H$ is*

$$H = \sum_{a=1}^{r} \lambda_a^H (v_a^H \otimes v_a^H \otimes v_a^H).$$

*Let $\Lambda^H = \mathrm{diag}(\lambda_1^H, \ldots, \lambda_r^H)$. Then the parameters of the mixture distribution are*

$$P = U_{M_2} \Sigma_{M_2}^{1/2} V^H \Lambda^H \quad and \quad Q = (\Lambda^H)^{-2} .$$

The main challenge in estimating $M_2$ (resp. $M_3$) from empirical data are the diagonal entires. In [12], alternating minimization approach is used for matrix completion to find the missing diagonal entries of $M_2$, and used a least squares method for estimating the tensor $H$ directly from the samples. Let $\Omega_2$ denote the set of off-diagonal indices for an $N \times N$ matrix and $\Omega_3$ denote the off-diagonal entries of an $N \times N \times N$ tensor such that the corresponding projections are defined as

$$\mathcal{P}_{\Omega_2}(M)_{ij} \equiv \begin{cases} M_{ij} & \text{if } i \neq j , \\ 0 & \text{otherwise} . \end{cases} \quad and \quad \mathcal{P}_{\Omega_3}(T)_{ijk} \equiv \begin{cases} T_{ijk} & \text{if } i \neq j, j \neq k, k \neq i , \\ 0 & \text{otherwise} . \end{cases}$$

for $M \in \mathbb{R}^{N \times N}$ and $T \in \mathbb{R}^{N \times N \times N}$.

In lieu of above discussion, we shall use $\mathcal{P}_{\Omega_2}(\hat{M}_2)$ and $\mathcal{P}_{\Omega_3}(\hat{M}_3)$ to obtain estimation of diagonal entries of $M_2$ and $M_3$ respectively. To keep technical arguments simple, we shall use first $|\mathcal{S}|/2$ samples based $\hat{M}_2$, denoted as $\hat{M}_2\big(1, \frac{|\mathcal{S}|}{2}\big)$ and second $|\mathcal{S}|/2$ samples based $\hat{M}_3$, denoted by $\hat{M}_3\big(\frac{|\mathcal{S}|}{2} + 1, |\mathcal{S}|\big)$ in Algorithm 2.

Next, we state correctness of Algorithm 2 when $\mu(M_2)$ is small; proof is in Appendix.

**Theorem 3.** *There exists universal, strictly positive constants $C, C' > 0$ such that for all $\varepsilon \in (0, C)$ and $\delta \in (0, 1)$, if*

$$|\mathcal{S}| \geq C' \frac{r N^4 \log(N/\delta)}{q_{\min} \sigma_1(M_2)^2 \varepsilon^2} \left( \frac{1}{\ell^2} + \frac{\sigma_1(M_2)}{\ell N} + \frac{r^4 \sigma_1(M_2)^4}{\sigma_r(M_2)^5} \right) , \quad and$$

$$N \geq C' r^{3.5} \mu^6 \left( \frac{\sigma_1(M_2)}{\sigma_r(M_2)} \right)^{4.5} ,$$

---

**Algorithm 2** SPECTRALDIST: Moment method for Mixture of Discrete Distribution [12]

1: **Input**: Samples $\{x_t\}_{t\in\mathcal{S}}$, number of types $r$, number of iterations $T$
2: $\tilde{M}_2 \leftarrow$ MATRIXALTMIN $\left( \hat{M}_2\big(1, \frac{|\mathcal{S}|}{2}\big), r, T \right)$               (see Algorithm 3)
3: Compute eigenvalue decomposition of $\tilde{M}_2 = \tilde{U}_{M_2}\tilde{\Sigma}_{M_2}\tilde{U}_{M_2}^T$
4: $\tilde{H} \leftarrow$ TENSORLS $\left( \hat{M}_3\big(\frac{|\mathcal{S}|}{2}+1, |\mathcal{S}|\big), \tilde{U}_{M_2}, \tilde{\Sigma}_{M_2} \right)$         (see Algorithm 4)
5: Compute rank-$r$ decomposition $\sum_{a\in[r]} \lambda_a^{\tilde{H}}(\hat{v}_a^{\tilde{H}} \otimes \hat{v}_a^{\tilde{H}} \otimes \hat{v}_a^{\tilde{H}})$ of $\tilde{H}$, using RTPM of [2]
6: **Output**: $\hat{P} = \tilde{U}_{M_2}\tilde{\Sigma}_{M_2}^{1/2}\hat{V}^{\tilde{H}}\hat{\Lambda}^{\tilde{H}}$, $\hat{Q} = (\hat{\Lambda}^{\tilde{H}})^{-2}$, where $\hat{V}^{\tilde{H}} = [\hat{v}_1^{\tilde{H}} \ \ldots \ \hat{v}_r^{\tilde{H}}]$ and $\hat{\Lambda}^{\tilde{H}} = \mathrm{diag}(\lambda_1^{\tilde{H}}, \ldots, \lambda_r^{\tilde{H}})$

---

*then there exists a permutation $\pi$ over $[r]$ such that Algorithm 2 achieves the following bounds with a choice of $T = C'\log(N\,|\mathcal{S}|)$ for all $i \in [r]$, with probability at least $1 - \delta$:*

$$|\hat{q}_{\pi_i} - q_i| \ \leq \ \varepsilon \,, \ \text{and} \quad \|\hat{P}_{\pi_i} - P_i\| \ \leq \ \varepsilon\sqrt{\frac{r\,q_{\max}\,\sigma_1(M_2)}{q_{\min}}}\,,$$

*where $\mu = \mu(M_2)$ defined in (8) with run-time $\mathrm{poly}(N, r, 1/q_{\min}, 1/\varepsilon, \log(1/\delta), \sigma_1(M_2)/\sigma_r(M_2))$.*

---

**Algorithm 3** MATRIXALTMIN: Alternating Minimization for Matrix Completion [12]

1: **Input**: $\hat{M}_2\big(1, \frac{|\mathcal{S}|}{2}\big), r, T$
2: Initialize $N \times r$ dimensional matrix $U_0 \leftarrow$ top-$r$ eigenvectors of $\mathcal{P}_{\Omega_2}(\hat{M}_2\big(1, \frac{|\mathcal{S}|}{2}\big))$
3: **for all** $\tau = 1$ to $T - 1$ **do**
4:    $\hat{U}_{\tau+1} = \arg\min_U \|\mathcal{P}_{\Omega_2}(\hat{M}_2\big(1, \frac{|\mathcal{S}|}{2}\big)) - \mathcal{P}_{\Omega_2}(UU_\tau^T)\|_F^2$
5:    $[U_{\tau+1}R_{\tau+1}] = \mathrm{QR}(\hat{U}_{\tau+1})$              (standard QR decomposition)
6: **end for**
7: **Output**: $\tilde{M}_2 = (\hat{U}_T)(U_{T-1})^T$

---

**Algorithm 4** TENSORLS: Least Squares method for Tensor Estimation [12]

1: **Input**: $\hat{M}_3\big(\frac{|\mathcal{S}|}{2}+1, |\mathcal{S}|\big), \hat{U}_{M_2}, \hat{\Sigma}_{M_2}$
2: Define operator $\hat{\nu} : \mathbb{R}^{r\times r\times r} \to \mathbb{R}^{N\times N\times N}$ as follows

$$\hat{\nu}_{ijk}(Z) = \begin{cases} \sum_{abc} Z_{abc}(\hat{U}_{M_2}\hat{\Sigma}_{M_2}^{1/2})_{ia}(\hat{U}_{M_2}\hat{\Sigma}_{M_2}^{1/2})_{jb}(\hat{U}_{M_2}\hat{\Sigma}_{M_2}^{1/2})_{kc}, & \text{if } i \neq j \neq k \neq i\,, \\ 0, & \text{otherwise.} \end{cases} \quad (13)$$

3: Define $\hat{A} : \mathbb{R}^{r\times r\times r} \to \mathbb{R}^{r\times r\times r}$ s.t. $\hat{A}(Z) = \hat{\nu}(Z)[\hat{U}_{M_2}\hat{\Sigma}_{M_2}^{-1/2}, \hat{U}_{M_2}\hat{\Sigma}_{M_2}^{-1/2}, \hat{U}_{M_2}\hat{\Sigma}_{M_2}^{-1/2}]$
4: **Output**: $\arg\min_Z \|\hat{A}(Z) - \mathcal{P}_{\Omega_3}\big(\hat{M}_3\big(\frac{|\mathcal{S}|}{2}+1, |\mathcal{S}|\big)\big)[\hat{U}_{M_2}\hat{\Sigma}_{M_2}^{-1/2}, \hat{U}_{M_2}\hat{\Sigma}_{M_2}^{-1/2}, \hat{U}_{M_2}\hat{\Sigma}_{M_2}^{-1/2}]\|_F^2$

---

### 3.2   Phase 2: RANKCENTRALITY.

Recall that $E = \{(i_k, j_k) : i_k \neq j_k \in [n], 1 \leq k \leq N\}$ represents collection of $N = |E|$ pairs and $G = ([n], E)$ is the corresponding graph. Let $\tilde{P}_a$ denote the estimation of $P_a = [P_{ka}] \in [-1, 1]^N$ for the mixture component $a, 1 \leq a \leq r$; where $P_{ka}$ is defined as per (1). For each $a$, using $G$ and $\tilde{P}_a$, we shall use the RANKCENTRALITY [19] to obtain estimation of $\mathbf{w}^{(a)}$. Next we describe the algorithm and guarantees associated with it.

Without loss of generality, we can assume that $\mathbf{w}^{(a)}$ is such that $\sum_i w_i^{(a)} = 1$ for all $a, 1 \leq a \leq r$. Given this normalization, RANKCENTRALITY estimates $\mathbf{w}^{(a)}$ as stationary distribution of an appropriate Markov chain on $G$. The transition probabilities are 0 for all $(i, j) \notin E$. For $(i, j) \in E$, they are function of $\tilde{P}_a$. Specifically, transition matrix $\tilde{p}^{(a)} = [\tilde{p}_{i,j}^{(a)}] \in [0, 1]^{n \times n}$ with $\tilde{p}_{i,j}^{(a)} = 0$ if

$(i,j) \notin E$, and for $(i_k, j_k) \in E$ for $1 \le k \le N$,

$$\tilde{p}_{i_k,j_k}^{(a)} = \frac{1}{d_{\max}} \frac{(1+\tilde{P}_{ka})}{2} \quad \text{and} \quad \tilde{p}_{j_k,i_k}^{(a)} = \frac{1}{d_{\max}} \frac{(1-\tilde{P}_{ka})}{2}, \tag{14}$$

Finally, $\tilde{p}_{i,i}^{(a)} = 1 - \sum_{j \neq i} \tilde{p}_{i,j}^{(a)}$ for all $i \in [n]$. Let $\tilde{\pi}^{(a)} = [\tilde{\pi}_i^{(a)}]$ be a stationary distribution of the Markov chain defined by $\tilde{p}^{(a)}$. That is,

$$\tilde{\pi}_i^{(a)} = \sum_j \tilde{p}_{ji}^{(a)} \tilde{\pi}_j^{(a)} \quad \text{for all } i \in [n]. \tag{15}$$

Computationally, we suggest obtaining estimation of $\tilde{\pi}$ by using power-iteration for $T$ iterations. As argued before, cf. [19], $T = \Theta\big(b^2 d_{\max}(\log n + \log(1/\varepsilon))/(\xi d_{\min})\big)$, is sufficient to obtain reasonably good estimation of $\tilde{\pi}$.

The underlying assumption here is that there is a unique stationary distribution, which is established by our result under the conditions of Theorem 1. Now $\tilde{p}$ is an approximation of the ideal transition probabilities, where $p^{(a)} = [p_{i,j}^{(a)}]$ where $p_{i,j}^{(a)} = 0$ if $(i,j) \notin E$ and $p_{i,j}^{(a)} \propto w_j^{(a)}/(w_i^{(a)} + w_j^{(a)})$ for all $(i,j) \in E$. Such an ideal Markov chain is *reversible* and as long as $G$ is connected (which is, in our case, by choice), the stationary distribution of this ideal chain is $\pi^{(a)} = \mathbf{w}^{(a)}$ (recall, we have assumed $\mathbf{w}^{(a)}$ to be normalized so that all its components up to 1).

Now $\tilde{p}^{(a)}$ is an approximation of such an ideal transition matrix $p^{(a)}$. In what follows, we state result about how this approximation error translates into the error between $\tilde{\pi}^{(a)}$ and $\mathbf{w}^{(a)}$. Recall that $b \equiv \max_{i,j \in [n]} w_i/w_j$, $d_{\max}$ and $d_{\min}$ are maximum and minimum vertex degrees of $G$ and $\xi$ as defined in (9).

**Theorem 4.** *Let $G = ([n], E)$ be non-bipartite and connected. Let $\|\tilde{p}^{(a)} - p^{(a)}\|_2 \le \varepsilon$ for some positive $\varepsilon \le (1/4)\xi b^{-5/2}(d_{\min}/d_{\max})$. Then, for some positive universal constant C,*

$$\frac{\|\tilde{\pi}^{(a)} - \mathbf{w}^{(a)}\|}{\|\mathbf{w}^{(a)}\|} \le \frac{C\, b^{5/2}}{\xi} \frac{d_{\max}}{d_{\min}} \varepsilon. \tag{16}$$

*And, starting from any initial condition, the power iteration manages to produce an estimate of $\tilde{\pi}^{(a)}$ within twice the above stated error bound in $T = \Theta\big(b^2 d_{\max}(\log n + \log(1/\varepsilon))/(\xi d_{\min})\big)$ iterations.*

Proof of the above result can be found in Appendix. For spectral expander (e.g. connected Erdos-Renyi graph with high probability), $\xi = \Theta(1)$ and therefore the bound is effectively $O(\varepsilon)$ for bounded dynamic range, i.e. $b = O(1)$.

## 4 Discussion

Learning distribution over permutations of $n$ objects from partial observation is fundamental to many domains. In this work, we have advanced understanding of this question by characterizing sufficient conditions and associated algorithm under which it is feasible to learn mixed MNL model in computationally and statistically efficient (polynomial in problem size) manner from partial/pair-wise comparisons. The conditions are natural – the mixture components should be "identifiable" given partial preference/comparison data – stated in terms of full rank and incoherence conditions of the second moment matrix. The algorithm allows learning of mixture components as long as number of mixture components scale $o(n^{2/7})$ for distribution over permutations of $n$ objects.

To the best of our knowledge, this work provides first such sufficient condition for learning mixed MNL model – a problem that has remained open in econometrics and statistics for a while, and more recently Machine learning. Our work nicely complements the impossibility results of [1].

Analytically, our work advances the recently popularized spectral/tensor approach for learning mixture model from lower order moments. Concretely, we provide means to learn the component even when only partial information about the sample is available unlike the prior works. To learn the model parameters, once we identify the moments associated with each mixture, we advance the result of [19] in its applicability. Spectral methods have also been applied to ranking in the context of assortment optimization in [5].

## Footnotes

[1]We shall assume that, outcomes of these $\ell$ pairs are independent of each other, but coming from the same MNL mixture component. This is effectively true even they were generated by first sampling a permutation from the chosen MNL mixture component, and then observing implication of this permutation for the specific $\ell$ pairs, as long as they are distinct due to the Irrelevance of Independent Alternative hypothesis of Luce that is satisfied by MNL.

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
