[Supplementary Material]

# Supplementary material for "Learning Mixed Multinomial Logit Model from Ordinal Data"

## A  Proof of Theorem 1

In order to apply Theorem 4, let $\Delta = p - \tilde{p}$, then

$$\|\Delta\|_2 \leq \|\Delta - \text{diag}(\Delta)\|_2 \ + \ \|\text{diag}(\Delta)\|_2 \ \leq \|\Delta - \text{diag}(\Delta)\|_F \ + \ \max_{i \in [n]} \Delta_{ii}$$

$$\leq \sqrt{\sum_{i \neq j} (p_{ij} - \tilde{p}_{ij})^2} \ + \ \max_{i \in [n]} \left| \sum_{j \neq i} (p_{ij} - \tilde{p}_{ij}) \right|$$

$$\leq \frac{1}{\sqrt{2}\, d_{\max}} \|P_a - \tilde{P}_a\| \ + \ \frac{1}{2\sqrt{d_{\max}}} \|P_a - \tilde{P}_a\| \ .$$

From Theorem 3, we know that $\|\Delta\|_2 \leq \frac{2}{\sqrt{d_{\max}}} \|P_a - \tilde{P}_a\| \leq \varepsilon \sqrt{(r q_{\max} \sigma_1(M_2))/q_{\min}}$ and substituting this into the bound in Theorem 4 we get the desired bound.

## B  Proof of the performance guarantee for the spectral method in Theorem 3

To simplify notations, we will assume that the indices of the output of the algorithm and the ground truths are matched such that the theorem holds with identity permutation. The spectral algorithm outputs $\hat{P} = \tilde{U}_{M_2} \tilde{\Sigma}_{M_2}^{1/2} \hat{V}^{\tilde{H}} \hat{\Lambda}^{\tilde{H}}$. From theorem 2, we know that $P = U_{M_2} \Sigma_{M_2}^{1/2} V^H \Lambda^H$, or equivalently $P = U_{M_2} \Sigma_{M_2}^{1/2} V^H Q^{-1/2}$. To show that $P$ and $\hat{P}$ are close, we would hope that each of the terms above to be close.

To that end, define

$$\widetilde{V} \ \equiv \ \tilde{\Sigma}_{M_2}^{-1/2} \tilde{U}_{M_2}^T P Q^{1/2} \ , \tag{17}$$

$$\widetilde{G} \ \equiv \ \sum_{i=1}^{r} \frac{1}{\sqrt{q_i}} (\tilde{v}_i \otimes \tilde{v}_i \otimes \tilde{v}_i) \ , \tag{18}$$

where $\widetilde{V} = [\tilde{v}_1 \ \tilde{v}_2 \ \dots \tilde{v}_r]$ (note that $\widetilde{G}$ is proxy of $V^H$). Now

$$\begin{aligned}
\left\| \hat{P} - P \right\|_2 \ &\leq \ \left\| \tilde{U}_{M_2} \tilde{U}_{M_2}^T P - P \right\|_2 + \left\| \hat{P} - \tilde{U}_{M_2} \tilde{U}_{M_2}^T P \right\|_2 \\
&= \ \left\| \tilde{U}_{M_2} \tilde{U}_{M_2}^T P - P \right\|_2 + \left\| \tilde{U}_{M_2} \tilde{\Sigma}_{M_2}^{1/2} \hat{V}^{\tilde{H}} \hat{\Lambda}^{\tilde{H}} - \tilde{U}_{M_2} \tilde{\Sigma}_{M_2}^{1/2} \widetilde{V} Q^{-1/2} \right\|_2 \quad \text{using (17)} \\
&\leq \ \left\| \tilde{U}_{M_2} \tilde{U}_{M_2}^T P - P \right\|_2 + \left\| \tilde{U}_{M_2} \tilde{\Sigma}_{M_2}^{1/2} (\widetilde{V} - \hat{V}^{\tilde{H}}) Q^{-1/2} \right\|_2 \\
&\quad + \left\| \tilde{U}_{M_2} \tilde{\Sigma}_{M_2}^{1/2} \hat{V}^{\tilde{H}} (Q^{-1/2} - \hat{\Lambda}^{\tilde{H}}) \right\|_2 \ . \tag{19}
\end{aligned}$$

To bound the three terms on the RHS of (19), we shall use the following 'errors' (which we shall bound sharply later in the proof): define (recall $\tilde{H}$ was produced by Algorithm Tensor Least Squares)

$$\varepsilon_M = \frac{\|M_2 - \tilde{M}_2\|_2}{\sigma_r(M_2)}$$

$$\varepsilon_H = \|\hat{H} - H\|_2. \tag{20}$$

**Bounding the first term in RHS of** (19). To begin with, note that we can represent $P = U_{M_2} \Sigma V^T$ by definition. Given the definition of (20), and an application of Davis-Kahan theorem [9] implies that

$$\|(\tilde{U}_{M_2} \tilde{U}_{M_2}^T - \mathbb{I}) U_{M_2}\|_2 \leq \varepsilon_M. \tag{21}$$

Using this, we have

$$\begin{aligned}
\|\tilde{U}_{M_2} \tilde{U}_{M_2}^T P - P\|_2 &= \|(\tilde{U}_{M_2} \tilde{U}_{M_2}^T - \mathbb{I}) P\|_2 \\
&\leq \|(\tilde{U}_{M_2} \tilde{U}_{M_2}^T - \mathbb{I})\|_2 U_{M_2} \|\Sigma V^T\|_2 \\
&\leq \varepsilon_M \|P\|_2 \ , \tag{22}
\end{aligned}$$

**Bounding the second term in RHS of** (19)**.** Consider

$$\|\tilde{\Sigma}_{M_2}\|_2 = \|\tilde{M}_2\|_2 \leq \|\tilde{M}_2 - M_2\|_2 + \|M_2\|_2$$
$$\leq \varepsilon_M \sigma_r(M_2) + \|M_2\|_2. \tag{23}$$

We state the following Lemma providing bound on $\|\widetilde{V} - \hat{V}^{\tilde{H}}\|_2$ in terms of $\varepsilon_M$ and $\varepsilon_G$:

**Lemma B.1.** *There exists universal constant $C_1 > 0$ such that*

$$\|\widetilde{V} - \hat{V}^{\tilde{H}}\|_2 \quad \leq \quad C_1 \sqrt{r\, q_{\max}} \left( \varepsilon_H + \frac{1}{\sqrt{q_{\min}}} \varepsilon_M \right), \; and$$

$$\|Q^{-1/2} - \hat{\Lambda}^H\|_2 \quad \leq \quad C_1 \left( \varepsilon_H + \frac{1}{\sqrt{q_{\min}}} \varepsilon_M \right).$$

Given (23), Lemma B.1, the fact that $\tilde{U}_{M_2}$ is a unitary matrix and $Q$ is a diagonal matrix, we obtain that the second term in RHS of (19) is bounded by

$$\left\| \tilde{U}_{M_2} \tilde{\Sigma}_{M_2}^{1/2} (\widetilde{V} - \hat{V}^{\tilde{H}}) Q^{-1/2} \right\|_2 \quad \leq \quad \left\| \tilde{U}_{M_2} \right\|_2 \left\| \tilde{\Sigma}_{M_2}^{1/2} \right\|_2 \left\| (\widetilde{V} - \hat{V}^{\tilde{H}}) \right\|_2 \left\| Q^{-1/2} \right\|_2$$

$$\leq \quad \frac{\sqrt{\varepsilon_M \sigma_r(M_2) + \|M_2\|_2}}{\sqrt{q_{\min}}} \|\widetilde{V} - \hat{V}^{\tilde{H}}\|_2$$

$$\leq \quad C_2 \sqrt{\frac{\|M_2\|_2\, r\, q_{\max}}{q_{\min}}} \left( \varepsilon_H + \frac{1}{\sqrt{q_{\min}}} \varepsilon_M \right), \tag{24}$$

for an appropriate universal constant $C_2 > 0$, with $\varepsilon_M \leq 1/2$ and using fact $\sigma_r(M_2) \leq \|M_2\|_2$.

**Bounding the third term in RHS of** (19)**.** Observe that

$$\|\hat{V}^{\tilde{H}}\|_2 \leq \|\widetilde{V}\|_2 + \|\hat{V}^{\tilde{H}} - \widetilde{V}\|_2$$
$$\leq 1 + 2\varepsilon_M + C_1 \sqrt{r q_{\max}}(\varepsilon_H + \varepsilon_M/\sqrt{q_{\min}}) \tag{25}$$

using Remark C.1 and Lemma B.1. Using (25) and Lemma B.1, the last term in (19) can be bounded above as

$$\left\| \tilde{U}_{M_2} \tilde{\Sigma}_{M_2}^{1/2} \hat{V}^{\tilde{H}} (Q^{-1/2} - \hat{\Lambda}^{\tilde{H}}) \right\|_2 \quad \leq \quad \left\| \tilde{U}_{M_2} \tilde{\Sigma}_{M_2}^{1/2} \right\|_2 \left\| \hat{V}^{\tilde{H}} \right\|_2 \left\| (Q^{-1/2} - \hat{\Lambda}^{\tilde{H}}) \right\|_2$$

$$\leq \left( \sqrt{\varepsilon_M \sigma_r(M_2) + \|M_2\|_2} \right) \|\hat{V}^{\tilde{H}}\|_2 \|Q^{-1/2} - \hat{\Lambda}^G\|_2$$

$$\leq C_3 \sqrt{\|M_2\|_2} \left( 1 + \sqrt{r q_{\max}}\varepsilon_H + \frac{\sqrt{r q_{\max}}}{\sqrt{q_{\min}}} \varepsilon_M \right) \left( \varepsilon_H + \frac{1}{\sqrt{q_{\min}}} \varepsilon_M \right) \tag{26}$$

$$\leq C_3 \sqrt{\|M_2\|_2} \left( \varepsilon_H + \frac{1}{\sqrt{q_{\min}}} \varepsilon_M \right), \tag{27}$$

for $\varepsilon_M \leq \sqrt{q_{\min}/(r\, q_{\max})}$, $\varepsilon_H \leq 1/\sqrt{r\, q_{\max}}$ and for some universal constant $C_3 > 0$.

**Towards Theorem 3.** Substituting (22), (24), (27) in (19), for universal constant $C_4 > 0$, we get

$$\|\hat{P} - P\|_2 \leq \varepsilon_M \|P\|_2 + C_4 \sqrt{\frac{\|M_2\|_2\, r\, q_{\max}}{q_{\min}}} \left( \varepsilon_H + \frac{1}{\sqrt{q_{\min}}} \varepsilon_M \right)$$

$$\leq C_4 \sqrt{\frac{\|M_2\|_2\, r\, q_{\max}}{q_{\min}}} \left( \varepsilon_H + \frac{1}{\sqrt{q_{\min}}} \varepsilon_M \right), \tag{28}$$

where we used the fact that $\|P\|_2 \leq \|\sqrt{\|M_2\|_2/q_{\min}}\|$ since $P = U_{M_2}\Sigma_{M_2}^{1/2}V^H\Lambda^H$. Given (28), to complete the proof of Theorem 3, we need to establish bounds on $\varepsilon_M$ and $\varepsilon_H$.

**Bounding $\varepsilon_M$.** To bound $\varepsilon_M$, we need to bound error between $M_2$ and $\tilde{M}_2$, the output of alternating minimization procedure applied to $\hat{M}_2$. The following theorem [12] provides such a bound.

**Theorem 5** (Theorem 4.1, [12])**.** *For an $N \times N$ symmetric rank-$r$ matrix $M$ with incoherence $\mu$, we observe off-diagonal entries corrupted by noise:*

$$\hat{M}_{ij} = \begin{cases} M_{ij} + E_{ij} & if\ i \neq j\,, \\ 0 & otherwise. \end{cases}$$

Let $\hat{M}^{(\tau)}$ denote the output after $\tau$ iterations of MATRIXALTMIN. If $\mu \leq (\sigma_r(M)/\sigma_1(M))\sqrt{N/(32\,r^{1.5})}$, the noise is bounded by $\|\mathcal{P}_{\Omega_2}(E)\|_2 \leq \sigma_r(M)/32\sqrt{r}$, and each column of the noise is bounded by $\|\mathcal{P}_{\Omega_2}(E)_i\| \leq \sigma_1(M)\mu\sqrt{3r/(8\,N)}$, $\forall i \in [N]$, then after $\tau \geq (1/2)\log\big(2\|M\|_F/\varepsilon\big)$ iterations of MATRIXALTMIN, the estimate $\hat{M}^{(\tau)}$ satisfies:

$$\|M - \hat{M}^{(\tau)}\|_2 \quad \leq \quad \varepsilon + \frac{9\,\|M\|_F\,\sqrt{r}}{\sigma_r(M)}\|\mathcal{P}_{\Omega_2}(E)\|_2 \,,$$

for any $\varepsilon \in (0,1)$. Further, $\hat{M}^{(\tau)}$ is $\mu_1$-incoherent with $\mu_1 \leq 6\mu\sigma_1(M_2)/\sigma_r(M_2)$.

To apply the above result in our setting to bound $\varepsilon_M$, we need to bound $\|\mathcal{P}_{\Omega_2}(E)\|_2$ and $\|\mathcal{P}_{\Omega_2}(E)_i\|$ for all $i \in [N]$. To that end, we state the following Lemma.

**Lemma B.2.** *Let $S_2 \equiv (2/|\mathcal{S}|) \sum_{t \in \{1,\ldots,|\mathcal{S}|/2\}} x_t x_t^T$ be the sample covariance matrix, and let $E = \frac{N(N-1)}{\ell(\ell-1)} S_2 - M_2$ denote the sampling error in the off-diagonal entries. Then, there exists a universal constant $C_5 > 0$ such that with probability at least $1 - \delta$,*

$$\|\mathcal{P}_{\Omega_2}(E)\|_2 \quad \leq \quad C_5 \sqrt{\frac{N^2 \log(N/\delta)}{\ell|\mathcal{S}|}\Big(\sigma_1(M_2) + \frac{N}{\ell}\Big)} \,.$$

*Moreover, the Euclidean norm of the columns are uniformly bounded by*

$$\|\mathcal{P}_{\Omega_2}(E)_i\| \quad \leq \quad C_5 \sqrt{\frac{N^3 \log(N/\delta)}{\ell^2|\mathcal{S}|}} \,,$$

*for all $i \in [N]$.*

Theorem 5 and Lemma B.2 imply that with probability at least $1 - \delta$, for large enough iterations of the MATRIXALTMIN,

$$\varepsilon_M = \frac{1}{\sigma_r(M_2)}\|M_2 - \tilde{M}_2\|_2 \leq C_6 \frac{\|M_2\|_F\,N}{\sigma_r(M_2)^2} \sqrt{\frac{r\,\log(N/\delta)}{\ell\,|\mathcal{S}|}\Big(\sigma_1(M_2) + \frac{N}{\ell}\Big)} \,, \qquad (29)$$

for universal constant $C_6, C_7 > 0$ when $|\mathcal{S}| \geq C_7 N^4 r \log(N/\delta)/(\ell^2 \sigma_r(M_2)^2)$ – the assumption of Theorem statement.

**Bounding $\varepsilon_H$.** To bound $\varepsilon_H$, we need bound on error induced by the output of the Tensor Least Square procedure. The following result [12] provides such a bound.

**Theorem 6** (Theorem 4.3, [12]). *If $N \geq \frac{144 r^3 \sigma_1(M_2)^2}{\sigma_r(M_2)^2}$, then with probability at least $1 - \delta$,*

$$\|H - \tilde{H}\|_F \quad \leq \quad \frac{24\mu_1^3\mu r^{3.5}\sigma_1(M_2)^{3/2}}{Nq_{\min}^{1/2}\sigma_r(M_2)^{3/2}}\varepsilon_M + 2\Big\|\mathcal{P}_{\Omega_3}\big(S_3 - M_3\big)\big[\tilde{U}_{M_2}\tilde{\Sigma}_{M_2}^{-1/2}, \tilde{U}_{M_2}\tilde{\Sigma}_{M_2}^{-1/2}, \tilde{U}_{M_2}\tilde{\Sigma}_{M_2}^{-1/2}\big]\Big\|_F \,,$$

*where $\varepsilon_M = (1/\sigma_r(M_2))\|\tilde{M}_2 - M_2\|$, $\mu = \mu(M_2)$, $\mu_1 = \mu(\tilde{U}_{M_2})$, and*

$$S_3 = \frac{N(N-1)(N-2)}{\ell(\ell-1)(\ell-2)} \frac{2}{|\mathcal{S}|}\Big(\sum_{t=1+|\mathcal{S}|/2}^{|\mathcal{S}|} x_t \otimes x_t \otimes x_t\Big). \qquad (30)$$

To utilize above result, we state the following Lemma.

**Lemma B.3.** *There exists a positive numerical constant $C_9$ such that with probability at least $1 - \delta$,*

$$\Big\|\mathcal{P}_{\Omega_3}\big(S_3 - M_3\big)\big[\tilde{U}_{M_2}\tilde{\Sigma}_{M_2}^{-1/2}, \tilde{U}_{M_2}\tilde{\Sigma}_{M_2}^{-1/2}, \tilde{U}_{M_2}\tilde{\Sigma}_{M_2}^{-1/2}\big]\Big\|_F \quad \leq \quad C_9 \frac{r^3\mu_1^3 N^{3/2}}{\sigma_r(M_2)^{3/2}}\sqrt{\frac{\log(1/\delta)}{|\mathcal{S}|}} \,.$$

Theorem 6 and Lemma B.3 imply that with probability at least $1 - \delta$

$$\varepsilon_H = \|H - \tilde{H}\|_2$$

$$\leq C_{10}\,\mu_1^3\,r^3\left(\frac{\mu r^{1/2}}{Nq_{\min}^{1/2}}\frac{\sigma_1(M_2)^{3/2}}{\sigma_r(M_2)^{3/2}}\varepsilon_M + \frac{N^{3/2}}{\sigma_r(M_2)^{3/2}}\sqrt{\frac{\log(1/\delta)}{|\mathcal{S}|}}\right) \qquad (31)$$

for some positive numerical constants $C_{10}, C_{11}$ when $N \geq C_{11} \frac{r^3 \sigma_1(M_2)^2}{\sigma_r(M_2)^2}$.

Equations (28)-(31) imply that

$$\|\hat{P} - P\|_2 \leq C_4 \sqrt{\frac{\|M_2\|_2 \, r \, q_{\max}}{q_{\min}}} \left( \varepsilon_H + \frac{1}{\sqrt{q_{\min}}} \varepsilon_M \right),$$

$$\leq C_4 \sqrt{\frac{\|M_2\|_2 \, r \, q_{\max}}{q_{\min}}} \left( C_{10} \, \mu_1^3 \, r^3 \left( \frac{\mu r^{1/2}}{N q_{\min}^{1/2}} \frac{\sigma_1(M_2)^{3/2}}{\sigma_r(M_2)^{3/2}} \varepsilon_M + \frac{N^{3/2}}{\sigma_r(M_2)^{3/2}} \sqrt{\frac{\log(1/\delta)}{|\mathcal{S}|}} \right) + \frac{1}{\sqrt{q_{\min}}} \varepsilon_M \right), .$$

Using $\mu_1 \leq 6\mu \sigma_1(M_2)/\sigma_r(M_2)$ from Theorem 5, we obtain (for appropriate constant $C_{12} > 0$),

$$\|\hat{P} - P\|_2 \leq C_{12} \sqrt{\frac{\|M_2\|_2 \, r \, q_{\max}}{q_{\min}^2}} \left\{ \left( \frac{\mu^4 r^{3.5}}{N} \left( \frac{\sigma_1(M_2)}{\sigma_r(M_2)} \right)^{4.5} + 1 \right) \varepsilon_M + \frac{\mu^3 r^3 \sigma_1(M_2)^3 N^{3/2} q_{\min}^{1/2}}{\sigma_r(M_2)^{4.5}} \sqrt{\frac{\log(1/\delta)}{|\mathcal{S}|}} \right\}.$$

$$\leq C_{12} \sqrt{\frac{\|M_2\|_2 \, r \, q_{\max}}{q_{\min}}} \left\{ \frac{2\varepsilon_M}{q_{\min}^{1/2}} + \mu^3 r^3 \frac{\sigma_1(M_2)^3 N^{1.5}}{\sigma_r(M_2)^{4.5}} \sqrt{\frac{\log(1/\delta)}{|\mathcal{S}|}} \right\}, \tag{32}$$

where $N \geq \mu^4 r^{3.5} (\sigma_1(M_2)/\sigma_r(M_2))^{4.5}$ as per assumption of the Theorem statement. From (29), it follows that when

$$|\mathcal{S}| \geq C_{13} \frac{r^2 \sigma_1(M_2)^2 N^2 (\sigma_1(M_2) + N/\ell) \log(N/\delta)}{\varepsilon^2 q_{\min} \ell \sigma_r(M_2)^4} \geq C_{13} \frac{r \|M_2\|_F^2 N^2 \log(N/\delta)}{\varepsilon^2 q_{\min} \ell \sigma_r(M_2)^4} \left( \sigma_1(M_2) + \frac{N}{\ell} \right), \tag{33}$$

(we used $\|M_2\|_F \leq \sqrt{r} \sigma_1(M_2)$ for rank $r$ matrix $M_2$) for an appropriate choice of universal constant $C_{13} > 0$,

$$\frac{2\varepsilon_M}{q_{\min}^{1/2}} \leq \frac{\varepsilon}{2}. \tag{34}$$

Also, recall that for (29) to hold, we require

$$|\mathcal{S}| \geq C_7 \frac{N^4 r \log(N/\delta)}{\ell^2 \sigma_r(M_2)^2}. \tag{35}$$

Also, when

$$|\mathcal{S}| \geq C_{14} \frac{\mu^6 r^6 N^3 \sigma_1(M_2)^6 \log(1/\delta)}{\varepsilon^2 \sigma_r(M_2)^9}, \tag{36}$$

for an appropriate choice of universal constant $C_{14} > 0$, the second term inside bracket in (32) is less than $\frac{\varepsilon}{2}$. From above, it follows that when

$$|\mathcal{S}| \geq C_{15} \frac{r^2 N^2 \sigma_1(M_2)^2 \log(N/\delta)}{q_{\min} \varepsilon^2 \sigma_r(M_2)^4} \left( \frac{N}{\ell^2} + \frac{\sigma_1(M_2)}{\ell} + \frac{q_{\min} \mu^6 r^4 \sigma_1(M_2)^4 N}{\sigma_r(M_2)^5} \right) + \frac{N^4 r \log(N/\delta)}{\ell^2 \sigma_1(M_2)^2} \tag{37}$$

$$\|\hat{P} - P\|_2 \leq \varepsilon \sqrt{\frac{r \, q_{\max} \sigma_1(M_2)}{q_{\min}}}, \text{ and}$$

$$\|Q^{-1/2} - \hat{\Lambda}^G\|_2 \leq \varepsilon,$$

for any $\varepsilon \in (0, C')$ for some positive constants $C = C_{15}$ and $C'$. Assuming $N \geq C'' r^{3.5} \mu^6 (\sigma_1(M_2)/\sigma_r(M_2))^{4.5}$, the above holds for

$$|\mathcal{S}| \geq C''' \frac{r N^4 \log(N/\delta)}{q_{\min} \sigma_1(M_2)^2 \varepsilon^2} \left( \frac{1}{\ell^2} + \frac{\sigma_1(M_2)}{\ell N} + \frac{r^4 \sigma_1(M_2)^4}{\sigma_r(M_2)^5} \right).$$

# C  Proof of the technical lemmas for the spectral method

## C.1  Proof of Lemma B.1

In order to apply the perturbation analysis of Theorem C.2 from [2], it is crucial that we compare to a tensor with an orthogonal decomposition. Since both $\tilde{H}$ and $\widetilde{G}$ do not have orthogonal decompositions, we define a new tensor $\bar{G}$ that is close to $\widetilde{G}$ and has an orthogonal decomposition. Given the singular value decomposition of $\widetilde{V} = XSY^T$, define $\bar{V} \equiv XY^T$. This $\bar{V} \in \mathbb{R}^{r \times r}$ is orthogonal such that $\bar{V}\bar{V}^T = \bar{V}^T\bar{V} = \mathbb{I}$, and is close to $\widetilde{V}$ such that

$$
\begin{aligned}
\|\widetilde{V} - \bar{V}\|_2 &= \|X(S - \mathbb{I})Y^T\|_2 \\
&\leq \max_{i \in [r]} |S_{ii} - 1| \\
&\leq 2\varepsilon_M \,,
\end{aligned}
\tag{38}
$$

where the last inequality follows from the next remark.

**Remark C.1** (Remark 10 in [12])**.** *Suppose* $\|M_2 - \tilde{M}_2\|_2 \leq \varepsilon_M \sigma_r(M_2)$, *then*

$$
\|\mathbb{I} - \widetilde{V}\widetilde{V}^T\|_2 \leq 2\varepsilon_M \,.
$$

It follows that $\|S^2 - \mathbb{I}\|_2 = \|\widetilde{V}\widetilde{V}^T - \mathbb{I}\|_2 \leq 2\varepsilon_M$. Therefore, $S_{ii}^2 \in [1 - 2\varepsilon_M, 1 + 2\varepsilon_M]$ and so is $S_{ii} \in [1 - 2\varepsilon_M, 1 + 2\varepsilon_M]$ for $\varepsilon_M \leq 1/2$.

Since $\|\widetilde{V} - \hat{V}^{\tilde{H}}\|_2 \leq \|\widetilde{V} - \bar{V}\|_2 + \|\bar{V} - \hat{V}^{\tilde{H}}\|_2 \leq 2\varepsilon_M + \|\bar{V} - \hat{V}^{\tilde{H}}\|_2$, we are left to show that $\|\bar{V} - \hat{V}^{\tilde{H}}\|_2 \leq 8\sqrt{r\, q_{\max}}(\varepsilon_H + (13/\sqrt{q_{\min}})\varepsilon_M)$ to finish the proof. Recall that $\widetilde{G} = \sum_{i=1}^r \frac{1}{\sqrt{q_i}}(\tilde{v}_i \otimes \tilde{v}_i \otimes \tilde{v}_i)$ and that $\hat{V}^{\tilde{H}}$ is the output of the robust power method applied to $\tilde{H}$, and let

$$
\bar{G} \equiv \sum_{i=1}^r \frac{1}{\sqrt{q_i}}(\bar{v}_i \otimes \bar{v}_i \otimes \bar{v}_i) \,.
$$

Applying (38), we get that

$$
\begin{aligned}
\|\widetilde{G} - \bar{G}\|_2 &= \max_{\|u\|=1} \left\| \sum_{i=1}^r \frac{1}{\sqrt{q_i}}(\tilde{v}_i \otimes \tilde{v}_i \otimes \tilde{v}_i - \bar{v}_i \otimes \bar{v}_i \otimes \bar{v}_i) \right\|_2 \\
&\leq \max_{\|u\|=1} \sum_{i=1}^r \frac{1}{\sqrt{q_i}}\left\{ (u^T\tilde{v}_i)^3 - (u^T\bar{v}_i)^3 \right\} \\
&\leq \max_{\|u\|=1} \sum_{i=1}^r \frac{1}{\sqrt{q_i}}\left( u^T(\widetilde{V} - \bar{V})e_i \right)\left( (u^T\tilde{v}_i)^2 + (u^T\tilde{v}_i)(u^T\bar{v}_i) + (u^T\bar{v}_i)^2 \right) \\
&\leq \frac{1}{\sqrt{q_{\min}}}\|\widetilde{V} - \bar{V}\|_2(3 + 6\varepsilon_M + \varepsilon_M^2) \\
&\leq \frac{13}{\sqrt{q_{\min}}}\varepsilon_M \,,
\end{aligned}
$$

where the last line holds for $\varepsilon_M \leq 1/2$. Since $\|\tilde{H} - \bar{G}\|_2 \leq \varepsilon_H + (13/\sqrt{q_{\min}})\varepsilon_M$, we show that $\hat{V}^{\tilde{H}}$ and $\bar{V}$ are close using the perturbation analysis for robust power method from [2].

**Theorem C.2** (Restatement of Theorem 5.1 by [2])**.** *Let* $G = \sum_{i \in [r]} \lambda_i(v_i \otimes v_i \otimes v_i) + E$, *where* $\|E\|_2 \leq C_1 \frac{\lambda_{min}}{r}$. *Then the tensor power-method after* $N \geq C_2(\log r + \log\log\left(\frac{\lambda_{max}}{\|E\|_2}\right)$, *generates vectors* $\hat{v}_i, 1 \leq i \leq r$, *and* $\hat{\lambda}_i, 1 \leq i \leq r$, *s.t.*,

$$
\|v_i - \hat{v}_{P(i)}\|_2 \leq 8\|E\|_2/\lambda_{P(i)}, \quad |\lambda_i - \hat{\lambda}_{P(i)}| \leq 5\|E\|_2.
\tag{39}
$$

*where* $P$ *is some permutation on* $[r]$.

Applying the above theorem to $\tilde{H}$ and $\bar{G}$, we get that $\|\hat{V}^{\tilde{H}} - \bar{V}\|_2 \leq 8\sqrt{r\,q_{\max}}(\varepsilon_H + (13/\sqrt{q_{\min}})\varepsilon_M)$. Notice that to apply the perturbation analysis, it is crucial that we use the fact that $\bar{G}$ has an orthogonal decomposition. Similarly, we can show that

$$\left|\hat{\lambda}_i - \frac{1}{\sqrt{q_i}}\right| \leq 5\varepsilon_H + \frac{65}{\sqrt{q_{\min}}}\varepsilon_M .$$

## C.2  Proof of Lemma B.2

Let $E = \mathcal{P}_{\Omega_2}(S_2 - \mathbb{E}[S_2]) = S_2 - \mathbb{E}[S_2] - \mathrm{diag}(S_2 - \mathbb{E}[S_2])$, and we bound each term separately using concentration inequalities. Define the random matrix $E^{(1)} \equiv S_2 - \mathbb{E}[S_2] = \frac{2}{|\mathcal{S}|}\sum_{t\in\{1,\dots,|\mathcal{S}|/2\}}\left(x_t x_t^T - \mathbb{E}[x_t x_t^T]\right)$. We apply the following matrix Bernstein bound for sum of independent sub-exponential random matrices with $X_t = x_t x_t^T - \mathbb{E}[x_t x_t^T]$.

**Theorem 7** (Theorem 6.2 of [24]). *Consider a finite sequence $\{X_t\}$ of independent, random, self-adjoint matrices with dimension $N$. Assume that*

$$\mathbb{E}[X_t] = 0 \quad and \quad \mathbb{E}[X_t^k] \preceq \frac{k!}{2}R^{k-2}A_t^2 \quad for\ k = 2,3,4,\dots$$

*Compute the variance parameter $\sigma^2 \equiv \|\sum_t A_t^2\|_2$. Then for all $a \geq 0$,*

$$\mathbb{P}\Big(\|\sum_t X_t\|_2 \geq a\Big) \leq N\exp\Big(\frac{-a^2/2}{\sigma^2 + Ra}\Big) .$$

The random matrix we defined $X_t$ is zero-mean, and satisfies $\mathbb{E}[X_t^k] \preceq \mathbb{E}[(x_t x_t^T)^k] = \ell^{k-1}\mathbb{E}[x_t x_t^T]$, which follows form the fact that $x_t^T x_t = \ell$ almost surely. We can prove the inequality $\mathbb{E}[X_t^k] \preceq \mathbb{E}[(x_t x_t^T)^k]$ via induction. Let $\bar{X} \equiv \mathbb{E}[x_t x_t^T]$. When $k = 1$, $\mathbb{E}[X_t] = 0 \preceq \mathbb{E}[(x_t x_t^T)]$, since $\bar{X}$ is a convex combination of positive semidefinite matrices. When $k = 2$, $\mathbb{E}[X_t^2] = \mathbb{E}[(x_t x_t^T)^2] - \bar{X}^2 \preceq \mathbb{E}[(x_t x_t^T)^2]$, since $\bar{X}$ is positive semidefinite. Suppose $0 \preceq \mathbb{E}[(x_t x_t^T)^k - X_t^k]$ for some $k \geq 1$, then this implies $\mathbb{E}[X_t\big((x_t x_t^T)^k - X_t^k\big)X_t] \succeq 0$. It follows that

$$\mathbb{E}\big[(x_t x_t^T)^{k+2} + \bar{X}(x_t x_t^T)^k\bar{X} - (x_t x_t^T)^{k+1}\bar{X} - \bar{X}(x_t x_t^T)^{k+1} - X_t^{k+2}\big] \succeq 0 ,$$

which implies

$$\begin{aligned}
\mathbb{E}\big[(x_t x_t^T)^{k+2} - X_t^{k+2}\big] &\succeq \mathbb{E}\big[(x_t x_t^T)^{k+1}\bar{X} + \bar{X}(x_t x_t^T)^{k+1} - \bar{X}(x_t x_t^T)^k\bar{X}\big] \\
&\succeq \ell^{k-1}\bar{X}(2\ell\mathbb{I} - \bar{X})\bar{X} \\
&\succeq 0 .
\end{aligned}$$

This follows from the fact that $\|\bar{X}\| \leq \mathbb{E}[\|x_t x_t^T\|] = \ell$. By induction, this proves the desired claim that $\mathbb{E}[X_t^k] \preceq \mathbb{E}[(x_t x_t^T)^k] = \ell^{k-1}\mathbb{E}[x_t x_t^T]$, for all $k$. Then, the condition in Theorem 7 is satisfied with $R = \ell$, $A_t^2 = \ell\,\mathbb{E}[x_t x_t^T]$, and $\sigma^2 = (|\mathcal{S}|/2)\,\ell\,\|\mathbb{E}[x_t x_t^T]\|_2$, which gives

$$\mathbb{P}\Big(\frac{|\mathcal{S}|}{2}\|E^{(1)}\|_2 \geq a\Big) \leq N\exp\Big(\frac{-a^2/2}{|\mathcal{S}|\ell\|\mathbb{E}[x_t x_t^T]\|_2/2 + \ell a}\Big) .$$

When $|\mathcal{S}| \geq (2\ell\log(N/\delta))/\|\mathbb{E}[x_t x_t^T]\|_2$ as per our assumption, the first term in the denominator dominates for $a = \sqrt{2|\mathcal{S}|\,\ell\,\|\mathbb{E}[x_t x_t^T]\|_2\,\log(N/\delta)}$. This implies that with probability at least $1 - \delta$,

$$\|E^{(1)}\|_2 \leq \sqrt{\frac{4\,\ell\,\|\mathbb{E}[x_t x_t^T]\|_2\,\log(N/\delta)}{|\mathcal{S}|}} ,$$

and since $\|\mathbb{E}[x_t x_t^T]\|_2 = \|\frac{\ell(\ell-1)}{N(N-1)}\mathcal{P}_{\Omega_2}(M_2) + (\ell/N)\mathbb{I}\|_2 \leq (\ell^2/N^2)\sigma_1(M_2) + (\ell/N)$, this gives the desired bound.

Now let $E^{(2)} \equiv \mathrm{diag}(S_2 - \mathbb{E}[S_2])$, where each diagonal term is distributed as binomial distribution $\mathrm{Binom}(|\mathcal{S}|/2, \ell/N)$. When $|\mathcal{S}| \geq (2N/\ell)\log(|\mathcal{S}|/\delta)$ standard Bernstein inequality gives

$$\|E^{(2)}\|_2 \leq \max_{i\in[N]}E^{(2)}_{ii} \leq \sqrt{\frac{4\ell\,\log(N/\delta)}{N\,|\mathcal{S}|}} ,$$

for all $i \in [N]$ with probability at least $1 - \delta$. Together we have the desired upper bound on $\|E^{(1)} + E^{(2)}\|_2$.

In the case of $\|E_i\|$, similar concentration of measure shows that $|E_{ij}| \leq \sqrt{\frac{8\,\ell^2\,\log(N/\delta)}{N^2\,|\mathcal{S}|}}$ with probability at least $1 - \delta$ for all $j \neq i$. This gives

$$\|\mathcal{P}_{\Omega_2}(E)_i\| \;\leq\; \sqrt{\frac{8\,\ell^2\,\log(N/\delta)}{N\,|\mathcal{S}|}}\;.$$

### C.3 Proof of Lemma B.3

Let $\hat{H}_{abc} = \frac{n(n-1)(n-2)}{\ell(\ell-1)(\ell-2)}\frac{2}{|\mathcal{S}|}\sum_{t=1+|\mathcal{S}|/2}^{|\mathcal{S}|}Y_{abc}^t$ where $Y_{abc}^t = \sum_{(i,j,k)\in\Omega_3} x_{t,i}x_{t,j}x_{t,k}\hat{Q}_{ia}\hat{Q}_{jb}\hat{Q}_{kc}$, and $\hat{Q} = \tilde{U}_{M_2}\tilde{\Sigma}_{M_2}^{-1/2}$. Then,

$$\begin{aligned}
Y_{abc}^t \;=\;& \langle x_t, \hat{Q}_a\rangle\langle x_t, \hat{Q}_b\rangle\langle x_t, \hat{Q}_c\rangle - \langle x_t, \hat{Q}_a\rangle\sum_{i\in[N]}(x_{t,i}^2\hat{Q}_{ib}\hat{Q}_{ic}) - \langle x_t, \hat{Q}_b\rangle\sum_i(x_{t,i}^2\hat{Q}_{ia}\hat{Q}_{ic}) \\
& - \langle x_t, \hat{Q}_c\rangle\sum_i(x_{t,i}^2\hat{Q}_{ia}\hat{Q}_{ib}) + 2\sum_{i\in[N]}x_{t,i}^3\hat{Q}_{ia}\hat{Q}_{ib}\hat{Q}_{ic}\;.
\end{aligned}$$

We claim that $|Y_{abc}^t| \leq 6\ell^3\mu_1^3 r^{3/2}N^{-3/2}(1 - \varepsilon_M)^{-3/2}\sigma_r(M_2)^{-3/2}$. Since $x_t$ has only $\ell$ non-zero entries and by incoherence of $\mu(\hat{M}_2) = \mu_1$, we get that $|\langle x_t, \hat{Q}_a\rangle| \leq \ell\mu_1\sqrt{r/(N(1-\varepsilon_M)\sigma_r(M_2))}$. Similarly, $|\sum_{i\in[N]}(x_{t,i}^2\hat{Q}_{ib}\hat{Q}_{ic})| \leq \ell\mu_1^2(r/N)(1 - \varepsilon_M)^{-1}\sigma_r(M_2)^{-1}$ and $|\sum_{i\in[N]}(x_{t,i}^3\hat{Q}_{ia}\hat{Q}_{ib}\hat{Q}_{ic})| \leq \ell\mu_1^3(r/N)^{3/2}(1 - \varepsilon_M)^{-3/2}\sigma_r(M_2)^{-3/2}$.

Applying Hoeffding's inequality to $\hat{H}_{abc}$, we get that

$$\left|\hat{H}_{abc} - H_{abc}\right| \;\leq\; \frac{48\,r^{3/2}\,\mu_1^3\,N^{3/2}}{\sigma_r(M_2)^{3/2}}\sqrt{\frac{\log(2/\delta)}{|\mathcal{S}|}}\;,$$

for $\varepsilon_M \leq 1/2$ with probability at least $1 - \delta$, where $H_{abc} = \mathcal{P}_{\Omega_3}(M_3)[\tilde{U}_{M_2}\tilde{\Sigma}_{M_2}^{-1/2}, [\tilde{U}_{M_2}\tilde{\Sigma}_{M_2}^{-1/2}, [\tilde{U}_{M_2}\tilde{\Sigma}_{M_2}^{-1/2}]$.

## D Proof of Theorem 4 for the error bound of RANKCENTRALITY

The proof builds on key technical Lemma from [19]. Recall that the comparison graph $G = ([n], E)$ has $N = |E|$ pairs/edges and the transition matrix for a random walk on this graph $G$ for mixture component $a, 1 \leq a \leq r$ is $\tilde{p}^{(a)} = [\tilde{p}_{i,j}^{(a)}] \in [0,1]^{n\times n}$ which depends on $\tilde{P}_a$, the estimation of $P_a$ obtained by the algorithm in phase 1.

The Markov chain $\tilde{p}^{(a)}$ is designed in such a way that if $\tilde{P}_a$ were indeed exactly equal to $P_a$, based on the true model parameters $\mathbf{w}^{(a)}$, then the following holds (easy to check): (a) Markov chain $\tilde{p}^{(a)}$ is irreducible and aperiodic as long as $G$ is connected, (b) the stationary distribution $\tilde{\pi}^{(a)}$ is equal (proportional) to $\mathbf{w}^{(a)}$ (without loss of generality, we assume that $\mathbf{w}^{(a)}$ are such that they sum up to 1). The above fact primarily holds because in this ideal scenario the correspond Markov chain is a reversible Markov chain with the desired stationary distribution. In reality, the Markov chain $\tilde{p}^{(a)}$ is an approximation of the ideal scenario and we need to quantity the error due to estimation error of $\tilde{P}_a$. This is precisely what we shall do next using the following Lemma of [19].

**Lemma D.1** (Lemma 2 in [19]). *For a Markov chain $\tilde{p}$ and an aperiodic, irreducible and reversible Markov chain $p$ with the stationary distribution $\pi$, let $\Delta = \tilde{p} - p$ and let $\tilde{\pi}^{(t)}$ be the distribution at time $t$ according to the Markov chain $\tilde{p}$ when started with initial distribution $\tilde{\pi}^{(0)}$. Then,*

$$\frac{\|\tilde{\pi}^{(t)} - \pi\|}{\|\pi\|} \;\leq\; \rho^t\frac{\|\tilde{\pi}^{(0)} - \pi\|}{\|\pi\|}\sqrt{\frac{\pi_{\max}}{\pi_{\min}}} + \frac{1}{1-\rho}\|\Delta\|_2\sqrt{\frac{\pi_{\max}}{\pi_{\min}}}\;,$$

*where $\pi_{\min} = \min_{i\in[n]}\pi_i$, $\pi_{\max} = \max_{i\in[n]}\pi_i$, $\rho = \lambda_{\max}(p) + \|\Delta\|_2\sqrt{\pi_{\max}/\pi_{\min}}$, and $\lambda_{\max}(p) = \max\{|\lambda_2(p)|,\ldots,|\lambda_n(p)|\}$ is the second largest eigenvalue of $p$ in absolute value.*

To prove the desired bound, we need control on the quantities $\|\Delta\|_2$ and $\rho$. We know that the spectral norm is bounded by $\|\Delta\|_2 \leq \varepsilon$ as per our assumption. The following lemma provides a lower bound on $1 - \rho$.

**Lemma D.2.** *For $\varepsilon \leq (1/4)\xi b^{-5/2}(d_{\min}/d_{\max})$, we have*

$$1 - \rho \ \geq \ \frac{1}{4}\frac{\xi}{b^2}\frac{d_{\min}}{d_{\max}} \ .$$

Note that we defined $b \equiv \pi_{\max}/\pi_{\min}$ and also it follows from the definition that $\|\pi\| \geq 1/\sqrt{n}$ and $\|\tilde{\pi}^{(0)} - \pi\| \leq 2$. Substituting the bounds on $\|\Delta\|_2 \leq \varepsilon$ and $1 - \rho$, we get that there exists positive numerical constant $C, C'$ such that for $t \geq C' \log\left(n/\varepsilon\right)/\log(\rho)$,

$$\frac{\|\tilde{\pi}^{(t)} - \pi\|}{\|\pi\|} \ \leq \ \frac{C\, b^{5/2}\, d_{\max}}{\xi\, d_{\min}}\varepsilon \ .$$

The necessary number of iterations can be further bounded by $t \geq C'(b^2 d_{\max}/(\xi d_{\min}))\left(\log(1/\varepsilon) + \log(n)\right)$ using Lemma D.2. This proves that the bound in Theorem 4, for $\varepsilon \leq (1/4)\xi b^{-5/2}(d_{\min}/d_{\max})$.

Now we are left to prove Lemma D.2. From the definition of $\rho$,

$$1 - \rho \ \geq \ 1 - \lambda_{\max}(p) - \varepsilon\sqrt{b} \ .$$

We will show that

$$1 - \lambda_{\max}(p) \ \geq \ \frac{\xi d_{\min}}{2b^2 d_{\max}} \ , \tag{40}$$

where the desired bound follows for $\varepsilon \leq (1/4)\xi d_{\min}/(b^{5/2}d_{\max})$ as per our assumption. To prove (40), we use the comparison theorems (cf. see [19]), which bound the spectral gap of the Markov chain $p$ of interest, by comparing it to a more tractable Markov chain. We use the simple random walk on the undirected graph $G([n], E)$ as a reference. Define the transition matrix of a simple random walk as

$$Q_{ij} \ = \ \begin{cases} \frac{1}{d_i} & \text{for } (i,j) \in E \ , \\ 0 & \text{otherwise} \ , \end{cases}$$

where $d_i$ is the degree of node $i$. The stationary distribution of this Markov chain is $\mu_i = d_i/\sum_j d_j$, and $Q$ is reversible since the detailed balance equation is satisfied, i.e. $\mu_i Q_{ij} = 1/(\sum_j d_j) = \mu_j Q_{ji}$ for all $(i,j) \in E$. The following key lemma provides a bound on the spectral gap of $p$ with respect to the spectral gap of $Q$.

**Lemma D.3** (Lemma 6 in [19]). *Let $Q, \mu$ and $p, \pi$ be reversible Markov chains on a finite set $[n]$ representing random walks on a graph $G([n], E)$, i.e. $p_{ij} = 0$ and $Q_{ij} = 0$ if $(i,j) \notin E$. For $\alpha \equiv \min_{(i,j)\in E}\{\pi_i p_{ij}/(\mu_i Q_{ij})\}$ and $\beta \equiv \max_{i\in\mathcal{V}}\{\pi_i/\mu_i\}$,*

$$\frac{1 - \lambda_{\max}(p)}{1 - \lambda_{\max}(Q)} \ \geq \ \frac{\alpha}{\beta} \ .$$

We have defined $\xi \equiv 1 - \lambda_{\max}(Q)$, and $\alpha$ and $\beta$ can be bounded as follows.

$$\begin{aligned}
\alpha \ &= \ \min_{(i,j)\in E} \frac{\pi_i p_{ij}}{\mu_i Q_{ij}} \\
&\geq \ \min_{(i,j)\in E} \frac{w_i w_j \sum_{k\in[n]} d_k}{d_{\max}(w_i + w_j)\sum_{k\in\mathcal{V}} w_k} \ , \ \text{and} \\
\beta \ &= \ \max_{i\in[n]} \frac{\pi_i}{\mu_i} \\
&\leq \ \max_{i\in[n]} \frac{w_i \sum_k d_k}{d_i \sum_k w_k} \ .
\end{aligned}$$

Hence, $\alpha/\beta \geq d_{\min}/(2d_{\max}b^2)$. This proves the desired bound in (40).

# E   Proof of Remark 2.1

Assuming $w_i^{(a)}$'s are drawn uniformly at random from the interval $[1, 2]$ and $G(\mathcal{V}, \mathcal{E})$ drawn from the Erdös-Rényi model with average degree $\bar{d} \geq \log n$, we want to bound the singular values and the incoherence of $M_2 = PQP^T$. Define a matrix $\tilde{M} = Q^{1/2}P^T P Q^{1/2} \in \mathbb{R}^{2 \times 2}$. Since $\tilde{M}$ and $M_2$ have the same set of non-zero singular values, we analyze the spectrum of $\tilde{M}$.

For our example, $\tilde{M} = \frac{1}{2}P^T P$. Define $P_1$ and $P_2$ be the two columns of $P$ such that $P = [P_1 \, P_2]$, then using McDiarmid's inequality we get that, conditioned on the graph $G$ with $N$ edges and maximum degree $d_{\max}$,

$$
\begin{aligned}
\left| P_1^T P_2 \right| &\leq \varepsilon N , \\
\left| \|P_1\|^2 - (\ln(3486784401/68719476736) + 3)N \right| &\leq \varepsilon N , \\
\left| \|P_2\|^2 - (\ln(3486784401/68719476736) + 3)N \right| &\leq \varepsilon N .
\end{aligned}
\tag{41}
$$

with high probability for any positive constant $\varepsilon > 0$. We provide a proof for $\|P_1\|^2$, and the others follow similarly. Conditioned on the graph $G$, $\|P_1\|^2 = \sum_{(i,j) \in \mathcal{E}} \left( \left(w_j^{(1)} - w_i^{(1)}\right)/\left(w_i^{(1)} + w_j^{(1)}\right) \right)^2 = f(w_1^{(1)}, \ldots, w_N^{(1)})$ is a function with bounded difference:

$$
\sup_{w_1^{(1)} \ldots, w_n^{(1)}, v_i^{(1)}} \left| f(w_1^{(1)}, \ldots, w_{i-1}^{(1)}, w_i^{(1)}, w_{i+1}^{(1)}, \ldots, w_n^{(1)}) - f(w_1^{(1)}, \ldots, w_{i-1}^{(1)}, v_i^{(1)}, w_{i+1}^{(1)}, \ldots, w_n^{(1)}) \right| \leq d_{\max} .
$$

It follows that

$$
\mathbb{P}\left( \left| \|P_1\|^2 - \mathbb{E}[\|P_1\|^2] \right| \geq \varepsilon N \,\Big|\, G \right) \leq 2\exp\left\{ -\frac{2\varepsilon^2 N^2}{d_{\max}^2 n} \right\} .
$$

For Erdös-Rényi random graphs with $\bar{d} \geq \log n$, we know that $d_{\max} = \Theta(\bar{d})$ and $N = \Theta(\bar{d}N)$ with high probability. Also, it is not too difficult to compute $\mathbb{E}[\|P_1\|^2] = \ln(3486784401/68719476736) + 3 \simeq 0.0189$. It follows that with high probability, (41) holds.

Given (41), we can decompose the matrix as

$$
\tilde{M} = \begin{bmatrix} (\ln(3486784401/68719476736) + 3)N & 0 \\ 0 & (\ln(3486784401/68719476736) + 3)N \end{bmatrix} + \Delta ,
$$

where $\|\Delta\|_2 \leq 2\varepsilon$. It follows that $\sigma_1(\tilde{M}) \leq 0.02N$ and $\sigma_2(\tilde{M}) \geq 0.017N$. Choosing $\varepsilon = 0.001N$, this proves the desired bound.

To bound the incoherence, consider the SVD of $M_2 = USU^T = PQP^T$. There exists a orthogonal matrix $R$ such that $US^{1/2} = PQ^{1/2}R$. Then, the $i$-th row of $U$ is $U_i = e_i^T PQ^{1/2}RS^{-1/2}$. We know, $Q = diag(1/2, 1/2)$, $S = \text{diag}(\sigma_1(M_2), \sigma_2(M_2))$, and $RR^T = R^T R = \mathbb{I}$. It follows that

$$
\begin{aligned}
\mu(M_2) &= \max_i \sqrt{N/2}\|U_i\| \\
&\leq \max_i \sqrt{N/2}\sqrt{P_{i1}^2 + P_{i2}^2}(1/\sqrt{2})\sqrt{1/\sigma_2(M_2)} \\
&\leq 15 .
\end{aligned}
$$