[Reviews · NeurIPS 2014]

Submitted by Assigned_Reviewer_30

Summary:

This paper extends the classic MultiNomial Logit (MNL) choice model to a general family of choice models named Mixed MNL, which can be seen as a parametric class of distributions over permutations (e.g., permutations of items according to user preference). The main contributions of the paper are (1) to identify sufficient conditions under which a mixed MNL can be learnt, and (2) to propose a two-phase algorithm to learn the proposed mixed MNL models in an efficient manner. Part of the interesting theoretical results shows that the model with r components can be learnt with sample size being polynomially in n (number of items of interest) and r (number of components).

Quality:

The problem choice modeling studied in this paper is a fundamental and critical problem to the social choice community, and the proposed model and algorithm for this problem are certainly of interest to the machine learning community. The theoretical results presented in this paper seem to be correct, and the interpretation of the results help readers to understand the results. So, overall, I think the quality of this paper is good.

Clarity:

The paper is clear and well written. I found that the problem is well explained, although the authors may want to introduce the maths for the MNL model before introducing the mixed MNL model. The related work are sufficient given the space limitation of the paper.

Originality:

The problem studied in this paper is to model users' choice behaviors, which has been widely studied in the social choice community. MNL (and multinomial probit) model has been studied by earlier literature with efficient algorithms available for learning and inference. Despite of the current advances, I think that the current extension of the MNL to the mixed MNL is interesting to the community, and the proposed theoretical results concerning the sufficient conditions for learning MNL are critical to the advances of choice modeling community. Overall, I found that this paper is innovative.

Significance:

The significance of this paper does not seem to be easy to judge based on the theoretical results, partially because the novelty of the model with few literature to compare with.

Questions/comments:
1. in a practical setting (e.g., choice modeling for the Sushi preference data (http://www.kamishima.net/sushi/)), what is the guideline for choosing the number of components r?
2. under what conditions, does the mixed MNL outperform the classic MNL?
3. the tensor factorization used in Phase 1 of the algorithm seems to be based on Tucker decomposition (TD). TD in practice can be slow, and I am wondering if the authors have considered the Parafac decomposition?
4. despite the excellent theoretical results presented in the paper, I think the paper will be more convincing if some results on practical problems (e.g., synthetic or real-world datasets) can be shown.
Summary: This paper extends the classic multinomial logit (MNL) model to the mixed MNL, and provides sufficient conditions concerning learning the mixed MNL model. Theoretical results show the sufficient conditions, and a two-phase algorithm has been proposed to learn the model parameters in the mixed MNL.

Submitted by Assigned_Reviewer_39

The paper exploits moment tensor approach to recover a mixture of multinomial logit model (MNL) from pairwise ranking data. Moment method (up to 3rd order moment tensor here) has been developed recently to recover latent parameters in some mixture models. This is the first time to apply such a method in finding hidden groups of voters in pairwise ranking. In the paper, the authors developed a polynomial complexity algorithm to learn mixed MNL using spectrum of the second and third order moments of rank statistics. Such an algorithm is provable to recover the parameters under certain incoherent conditions, inherited from learning general mixture models.

The paper is mathematically solid and well-written. Up to the reviewer's knowledge, no one has applied the moment tensor method in the study of pairwise ranking problem. One shortcome lies in that tensor moment method has been critized for its instability. So in such an application study in ranking, it would be better to explore some numerical experiments rather than merely applying theoretical results from general problems.
Summary: The paper applies moment tensor method in identifying mixture models to the setting of pairwise ranking with mixture of multinomial logit models, establishing specific theoretical conditions for consistency.

Submitted by Assigned_Reviewer_42

This paper studies a rank data model as a mixture of Luce (or Plackett-Luce) Model. They provide conditions for estimating parameters of this model along with a moment-based algorithm. They further relate their theory to the algorithm.

This paper has nice technical results and they are presented well. I have a couple of concerns/suggestions with the following pieces of the presentation:

1- The authors state that they are learning Mixture Multinomial Logit models. Despite being closely related, their model is only considering Mixture of Luce model. The concerning differences are as follows:
i) MNL originally and traditionally, is used when you also observe alternative characteristics (also named as Multinomial logit regression) and the significance of research on Mixed MNL and MNL are due to the econometrics application. However, this paper does not get into that aspect and does not relate the results to an econometric setting.

ii) In most of applications, MNL is used for choice data, where we observe top 1 choice of agents (in fact their market shares). However, the theories and algorithms in this paper do not talk about this case. In fact, this paper only concerns pairwise data either observed directly as independent pairs or after the pre-process from full rankings.

Therefore, I suggest clarifying the beam of this paper and its implications in MNL and MMNL.

2- Since theorem 3 guarantees the correctness with high probability, isn’t the correctness of the algorithm an approximate correctness? Also, a condition (ASSUMPTION 1) in David Hunter’s paper (MM algorithms for generalized Bradley-Terry models. In The Annals of Statistics , volume 32, pages 384–406, 2004) depicts that there are cases that can lead to irreducibility of the defined graph, with the data.

From Hunter et al.: "As noted by Ford (1957), if it is possible to partition the set of individuals [alternatives] into two groups A and B such that there are never any inter group comparisons, then there is no basis for rating any individual in A with respect to any individual in B. On the other hand, if all the intergroup comparisons are won by an individual from the same group, say group A, then if all parameters belonging to A are doubled and the resulting vector renormalized, the likelihood must increase; thus, the likelihood has no maximizer. The following assumption [Ford (1957)] eliminates these possibilities."

And I wonder if Hunter's assumption 1 is not needed for the correctness of the proposed algorithms?

3- I can’t make sense of proposition 2.1

4- There are no experiments comparing the proposed algorithm with former works and there are no experiments describing the performance which is claimed theoretically. There is a large body of work and algorithms for Mixed Multinomial Logit model (as the authors indicated in introduction) and I really liked to see a comparison with some real data or a synthetic setting.

The problem and results of the paper are presented clearly.

1- As the authors indicate, there is a lot of work done in this line of research and there are some papers which algorithmically address the exact same estimation problem as this paper and in terms of application it is not clear what the gain of the new algorithm is. For example, the following papers proposed algorithms for estimating mixture of placket-Luce Model:

A mixture of experts model for rank data with applications in election studies
Authors: Isobel Claire Gormley, Thomas Brendan Murphy

Clustering ranked preference data using sociodemographic covariates
Authors: Isobel Claire Gormley, Thomas Brendan Murphy

Generalized Random Utility Models with Multiple Types
Authors: Hossein A. Soufiani, Hansheng Diao , Zhenyu Lai, David C. Parkes,

A Markov Chain Approximation to Choice Modeling
Authors: Jose Blanchet, Guillermo Gallego, Vineet Goyal

2- The results on identifying conditions for the mixture of PL models is original and interesting. Authors mention their results as “identifiable” in the discussion. However, it is not clear whether this is a formal statistical identifiablity claim.

3- Combining moment-based methods for ranks in order to find efficient estimators is interesting, and have been used in the following paper: Azari et al., Computing Parametric Ranking Models via Rank-Breaking and same authors, Generalized Method-of-Moments for Rank Aggregation. It is interesting to see how this work relates and compares to their methods experimentally and theoretically.

The interest in this work can be two folds, from an operations research aspect or rank aggregation aspect.
From the operation research point of view, this work lacks clear connection between the claims and an application through computational experiments to illustrate the implications of the new model for the specific OR problem.

From the rank aggregation aspect, and viewing the model as mixture of PL or BTL, the conditions and results are interesting, and with clear presentation this is an original contribution.

Summary: This paper is addressing an interesting and important problem with mixture of PL model for rank data. With addition of some experimental results and comparisons, the merits of this paper can be more clear, otherwise it is hard to judge the significance of the new algorithm.
Author Feedback
Author rebuttal: We would like to thank reviewers for detailed reading and constructive feedback. We start by summarizing, in our opinion, the primary contributions of this work followed by answers to specific reviewers’ comments.

The question of learning Mixed MNL (or PL) choice model from partial preference observations (e.g. pairwise comparisons) has been of interest in Operations Research, Econometrics/Political Science, and more recently Machine Learning coming from diverse sets of applications. Despite the long standing interest, there has not been any significant result providing characterization of conditions under which such a model can be learnt both statistically and computationally efficiently. In this work, we address this problem by identifying non-trivial conditions under which model can be learnt using polynomially (in problem size) many samples via a polynomial algorithm. We hope that this theoretical work will find applications across domains mentioned above.

Responses to Reviewers 30 and 39:

1. How to choose r ?

A simple, popular and data-driven approach is to iterate over various choices of r (num of components) and choose the one that fits the data best. In general, it’s an active research area. For example, in the context of Matrix factorization, the selection of rank can be done using algorithm in “OptSpace: A Gradient Descent Algorithm on Grassmann Manifold for Matrix Completion” by Keshavan and Oh.

2. When does MMNL outperform MNL?

By definition, MNL is a special case of MMNL, and hence MMNL model works at least as well as the MNL. It’s likely to outperform MNL when the true preference distribution is “multimodal” rather than “unimodal”, i.e. population is diverse in it’s preferences.

3. Is your algorithm faster than TD ?

The Phase 1 of the proposed approach uses CP (CANDECOMP/PARAFAC) decomposition, which is faster than Tucker decomposition (TD), mainly due to the fact that the structure of the third moment tensor has an underlying CP factorization.

4. What about experiments?

The primary contribution of this work is theoretical and goal was to characterize the conditions under which moment based approach is likely to provide sufficient conditions for learning MMNL. An extensive experimental validation based on real data is an important next step that we hope to pursue.

Response to Reviewer 42:

1.i) Plackett-Luce (PL) vs MNL?

Both PL as well as MNL model are identical in terms of inducing marginal distributions over pairwise comparisons if their underlying parameters are the same. And this paper primarily focusses on learning these parameters (and hence models) from pairwise comparisons. Therefore, our work can be viewed as either learning mixture of MNL (i.e. MMNL) or mixture of PL model.

1.ii) What about top choice data?

Reviewer is absolutely right that there are many applications (e.g. customer bought A when s/he observed A, B, C and D) where only observations are top (few) choices. However, they naturally translate into pair-wise comparisons (here, A > B, A > C, A > D). Of course, these are not “randomly” chosen comparisons. However, our model is an important (if we may say so) step towards such generality. It should be noted that there are natural applications where pairwise comparisons for specific pairs of items can be seeked (e.g. polls/surveys/crowdsourcing).

2. Is the guarantee approximate?

Theorem 3 gives approximation guarantees, and we will restate that we prove `approximation guarantees of Algorithm 2’. Regarding Hunter’s irreducibility condition, one implication of our theorem is that under the conditions of our main results, such irreducibility in the corresponding comparisons graph does not occur (with high probability).

3. Can’t make sense of Proposition 2.1?

It will be rephrased as “When $\overline{d}\geq \log n$ in the above example, we have $\sigma_1(M_2)\leq 0.02N,\sigma_2(M_2)\geq0.017N$, and $\mu(M_2)\leq 15$ with high probability.”

4. What about experiments?

As mentioned earlier, the primary purpose of this work is theoretical; experimental validation is the next step for this work.

Re. next sets of remarks.

1. Comparisons to existing work?

The main contribution of this work, which sets this work apart from the existing literature (and in particular references mentioned) is that we propose a new algorithm and provide finite sample guarantees for learning mixture of MNL/PL model. This has not been done previously and has been an important open problem. For example, the algorithm of Blanchet et al. will not find the correct Mixture of PL model even with infinite samples. The guarantees of Soufiani et al. is only on the asymptotic identifiability of the model itself and not about the algorithm and certainly not for finite samples. Gormley et al.’s papers have no theoretical guarantees to compare.

2. Is the model identifiable?

We mean that the conditions assumed in our theorem are sufficient for the model to be identifiable, i.e. given infinite number of samples precise inference of the model is possible. However, we do not mean that the conditions are necessary for identifiability.

3. We would like to thank the reviewer for the references, and these references will be added to the manuscript.